# Quantifying the interplay of Sea Ice Meltwater and Ice—Albedo Feedbacks in the Arctic Ice-Ocean System

Haohao Zhang[1, 2, 3], Andrea Storto[2], Xuezhi Bai[1, 3] and Chunxue Yang[2]

[1]College of Oceanography, Hohai University, Nanjing 210024, China.
[2]Institute of Marine Sciences (ISMAR), National Research Council (CNR), Rome, Italy.
[3]Key Laboratory of Marine Hazards Forecasting, Ministry of Natural Resources, Hohai University, Nanjing 210024, China.

*Correspondence to:* Xuezhi Bai (xuezhi.bai@hhu.edu.cn)

**Abstract.** Sea ice melting generates multiple feedbacks through the release of sea-ice meltwater (SIMW) and the expansion of open water. Due to the tight coupling of the ice-ocean system, these feedbacks are challenging to quantify independently. We employ a one-dimensional coupled sea ice-ocean model, by removing SIMW or keeping sea ice constant during the melting season, to quantify the independent effects of SIMW and ice-albedo feedbacks on the Arctic ice-ocean system. The experiments reveal the following: (1) The strong stratification induced by SIMW traps a portion of solar radiation, forming the Near-Surface Temperature Maximum (NSTM) and generating a negative feedback that reduces summer ice melting by 19% (feedback factor $\gamma$ = -0.19). (2) The ice-albedo positive feedback amplifies summer ice melting by 41% ($\gamma$ = +0.41). (3) These two feedbacks exhibit nonlinear interdependence: disabling the ice-albedo feedback reduces the SIMW feedback strength to -0.09, while eliminating SIMW effects enhances the ice-albedo feedback to +0.46. The impact of summer melting extends into the freezing season: intensified summer ice melt enhances early winter ice formation in strongly stratified regions, as reduced ice thickness and expanded open water areas in early winter accelerate oceanic heat loss, thereby promoting rapid freezing. Conversely, in the weakly stratified western Nansen Basin, summer SIMW release plays a critical role in preventing the upward mixing of Atlantic Warm Water, thereby protecting the winter ice cover. The SIMW influences are more pronounced under thinner initial sea ice, indicating that as Arctic sea ice continues to decline and Atlantification intensifies, the role of SIMW in regulating the Arctic ice-ocean system is expected to become increasingly significant.

## 1 Introduction

The vertical structure of the Arctic Ocean is characterized by a cold and fresh surface mixed layer overlying a cold halocline layer and the Atlantic warm water (AWW) layer at depth. Changes of the mixed layer and cold halocline layer collectively influence the upper ocean stratification in the Arctic Ocean, thereby significantly affecting heat fluxes from the deep-water layer to the surface and ice bottom (Aagaard et al., 1981; Rudels et al., 1996; Steele and Boyd, 1998). The large amount of freshwater in the upper Arctic Ocean is a key factor in the formation of its strong stratification structure (Carmack et al., 2016; Forryan et al., 2019; Haine et al., 2015).

On the interannual scale, the freshwater input from meteoric water (e.g., net precipitation and river runoff) governs the freshwater balance and stratification of the Arctic Ocean (Serreze et al., 2006), as it significantly contributes to the freshwater content-equivalent to a freshwater layer approximately 10 meters in the upper Arctic Ocean (Bauch et al., 1995; Dodd et al., 2012). On the seasonal scale, the fresh surface mixed layer is largely influenced by the sea ice melting/freezing cycle (Hordoir et al., 2022; Morison and Smith, 1981; Peralta-Ferriz and Woodgate, 2015; Polyakov et al., 2013). Each summer, sea-ice melt contributes approximately 1.2 meters of freshwater to the upper Arctic Ocean (Haine et al., 2015), which is markedly less than the long-term meteoric freshwater inventory of ~10 meters. However, this input is released intensively over months (~11,300 km³ during summer) (Haine et al., 2015), which can result in relatively thin sea-ice meltwater (SIMW) layers in the upper ocean and rapidly establish a shallow summer mixed layer (Hordoir et al., 2022; Peralta-Ferriz and Woodgate, 2015; Smith et al., 2023). Although river runoff also contributes a substantial freshwater during summer to the Arctic Ocean (~4,200 km³), it tends to remain confined to the coastal regions on seasonal scales (Osadchiev et al., 2020, 2021), with its subsequent transport pathways into the deep basin influenced by atmospheric circulation regimes (Wang et al., 2021a).

Over the past few decades, the extent of Arctic sea ice has shown significant negative trends characterized by a regime shift from multi-year ice to seasonal ice dominance (Kwok, 2018; Serreze and Meier, 2019; Sumata et al., 2023), indicating that the seasonal melt-freeze cycle is intensifying. Recent observational studies have highlighted the prevalence and importance of small-scale SIMW features during summer (Salganik et al., 2023a; Smith et al., 2023). In particular, thin (on the order of 0.1 to 1.0

m) under-ice SIMW layers and the subsequent formation of 'false bottoms' (new ice layers at the SIMW-seawater interface) have been identified as an important process in the ice-ocean system (Salganik et al., 2023a; Smith et al., 2023). For instance, observations from the MOSAiC expedition indicate that these 'false bottoms' can cover approximately 20% of the under-ice area and reduce bottom ice melt by 7–8% by insulating sea ice from oceanic heat (Salganik et al., 2023a). The shallow summer mixed layer formed by the SIMW also isolates a portion of the solar radiation heat entering the ocean into the base of the mixed layer, thereby forming the Near-Surface Temperature Maximum (NSTM) (Hudson et al., 2013; Jackson et al., 2010), which helps to slow down the ice melting during summer (Perovich et al., 2021; Zhang et al., 2023). Observations in the Canada Basin have revealed a significant shoaling trend of the NSTM, with its mean depth shoaling from more than 30 m in the late 1990s to approximately 20 m in the 21st century, largely driven by enhanced surface stratification resulting from accelerated sea ice melt (Gallaher et al., 2017; Jackson et al., 2010; Steele et al., 2011). In winter, the upward mixing of heat from subsurface and deep layers to mixed layer (e.g., the NSTM and AWW) can significantly impede sea ice growth (Smith et al., 2018; Steele et al., 2011; Timmermans et al., 2017). In particular, brine rejection during sea ice formation is an important driver of this upward heat entrainment: the release of dense, saline water caused by sea water freezing enhances vertical convection, deepens the mixed layer, and draws oceanic heat into the surface layer (Polyakov et al., 2020; Zhong et al., 2024). This heat flux usually does not stop ice formation, otherwise there would be no convective mixing and ongoing heat release to the mixed layer; instead, this heat flux represents a negative feedback mechanism that reduces the rate of winter ice growth (Polyakov et al., 2013), known as the ice production-entrainment feedback (Goosse et al., 2018).

Sea ice melting not only influences the ice-ocean system by altering ocean surface freshwater flux, but also leads to the expansion of open-water areas, which also generates significant feedback effects on ocean stratification and the ice itself by altering ocean vertical heat transport (Fine et al., 2023; Landrum and Holland, 2022; Polyakov et al., 2017, 2020). On the one hand, the reduction in sea ice cover facilitates increased momentum transfer from the atmosphere to the Arctic Ocean, resulting in enhanced vertical mixing (Armitage et al., 2020; Krishfield et al., 2014; Martin et al., 2016; Wang et al., 2024). On the other hand, it allows greater penetration of solar radiation into the ocean, contributing to further melting of sea

ice (Himmich et al., 2024; Holland et al., 2006; Jenkins and Dai, 2021). Consequently, any change in the melt-freeze cycle and the strength of stratification can have significant impacts on each other, through a complex series of feedback processes in the ice-ocean system.

Under the current climate conditions of rapid Arctic warming and intensifying sea ice melt-freeze cycles, separate quantification of the feedbacks induced by SIMW release and ice loss are crucial for understanding the coupling mechanisms and the relative importance of the various components of the Arctic ice-ocean system, as well as for predicting future climate change. However, the SIMW and ice-albedo feedbacks, while strongly coupled, exhibit distinct physical mechanisms and impacts: the SIMW feedback primarily arises from the freshwater input during sea ice melting, which enhances ocean stratification and decreases the ocean-to-ice heat flux, and can suppress further ice melting (a negative feedback). This process also influences winter ice formation by modulating heat entrainment (Zhang et al., 2023). In contrast, the ice-albedo feedback stems from the reduction in sea ice cover, which lowers surface albedo, increases solar radiation absorption, and amplifies ice melting (a positive feedback).

Due to the nearly inseparable coupling within the ice-ocean system and the strongly non-linear nature of these feedbacks, conducting an isolated quantitative study is challenging. Indeed, there are only a few studies that specifically quantify the impact of SIMW in the Arctic ice-ocean system. Zhang et al. (2023) demonstrated that the removal of SIMW in a coupled sea ice-ocean model increases ice melt by 17%. As meltwater removal weakens ocean stratification, allowing more heat from NSTM to reach the ice base, thereby accelerating melting. However, it remains largely unknown how the importance of the SIMW feedback compares to that of the ice-albedo feedback, which is known to have a profound impact on the Arctic Ocean. Moreover, the intrinsic components of each feedback—such as their independent strengths and nonlinear interactions—are poorly understood. To complicate the question, the regional differences in the vertical structure of the Arctic Ocean—characterized by a gradual weakening of stratification strength from the Amerasian Basin to the Eurasian Basin (Polyakov et al., 2013; Toole et al., 2010)—lead to spatially varying impact of SIMW on the ice-ocean system. To address these gaps mentioned above, this study employs a modeling approach to disentangle and quantify the effects of SIMW and ice-albedo feedbacks. Specifically, we aim to:

(1) Quantify the independent effects of SIMW and ice-albedo feedbacks on summer ice melting;

(2) Assess how the strength of each feedback changes when the other is suppressed, and their

nonlinear interdependence;

(3) Explore regional variations in the impact of SIMW on the ice-ocean system, considering

stratification differences.

To study these three questions, we used an idealized one-dimensional (1D) coupled sea ice-ocean

model, employing the methods of either removing the SIMW flux at the ocean surface or maintaining a

constant ice cover during the melting season, fully or partially decoupling the interactions between sea

ice and the ocean in the model, thereby preventing the independent effects of the SIMW and ice-albedo

feedbacks. For various feedbacks in polar regions, Goosse et al. (2018) proposed a simple and consistent

method to quantify these feedbacks, and we combine this approach with our model experiments to

quantify these two feedbacks. 1D models have been widely used in previous studies of the Arctic Ocean's

vertical structure and ice cover (Davis et al., 2016; Linders and Björk, 2013; Nummelin et al., 2015; Toole

et al., 2010; Wang et al., 2024). When advection flux effects are not considered, 1D model is a simple

and effective tool that can simulate a reasonable upper ocean stratification in short-term simulations. To

the best of our knowledge, this is the first study to quantify the above feedback effects based on a coupled

ice-ocean model, both individually and in combination, while also examining the varying initial

stratifications across different Arctic regions, to assess their relative impacts throughout the annual cycle.

The structure of this paper is as follows: Section 2 details the 1D model and experimental design.

Section 3 presents the model results. Section 4 calculates the feedback factors. Section 5 discusses the

results. Section 6 summarizes the main achievements. In the Supporting Information (SI), we provide

additional text and figures detailing the validation and performance of the 1D model.

**2 Materials and Methods**

**2.1 One-dimensional model**

The model used in this study is a 1D coupled sea ice-ocean model based on the Massachusetts Institute

of Technology General Circulation Model (MITgcm, Marshall et al., 1997). The ocean model utilizes the

nonlinear equation of state of Jackett and Mcdougall, (1995) and the vertical mixing is parameterized by

the nonlocal K-Profile Parameterization (KPP) scheme of Large et al., (1994) (see Section 1 of the SI for

more details regarding the model description and equations). The background vertical diffusivity in this model is set to $5.44 \times 10^{-7} \ m^2 s^{-1}$, following Nguyen et al., (2011) in their Arctic regional model based on MITgcm. Several observation-based studies have shown that the vertical diffusivity in the deep central Canadian Basin averages near-molecular levels, between $2.2 \times 10^{-7} \ m^2 s^{-1}$ and $3.4 \times 10^{-7} \ m^2 s^{-1}$ (Shaw and Stanton, 2014) and it averages $10^{-6} \ m^2 s^{-1}$ in the deep interior basins (Fer, 2009; Rainville and Winsor, 2008). Therefore, the choice of background vertical diffusivity is not uniform in various models, ranging from $10^{-7} \ m^2 s^{-1}$ to $10^{-6} \ m^2 s^{-1}$ (Liang and Losch, 2018; Linders and Björk, 2013; Nummelin et al., 2015; Toole et al., 2010). To check the uncertainty in our experimental results due to the choice of background mixing coefficients, we conducted experiments using the three coefficients of $10^{-7} \ m^2 s^{-1}$, $5.44 \times 10^{-7} \ m^2 s^{-1}$ and $10^{-6} \ m^2 s^{-1}$ and compared their differences. The results showed that selecting any of these three mixing coefficients did not significantly affect our experimental results. The differences between the experimental results using different mixing coefficients are detailed in Figure S1-S2 in the SI.

The water column in the model extends from the surface down to a depth of 700 m, with the vertical grid maintaining a uniform thickness of 1 m. At the bottom, a closed boundary condition is applied, meaning that there is no exchange of mass, momentum, or other properties across this boundary. We also verified that adding a sponge layer with a 1-day recovery period in the bottom 100 m results in no significant difference in the results compared to the case without any bottom boundary conditions (Figure S3-S4 in the SI), because the depth of the model is deep enough. The sea ice model is based on a variant of the viscous-plastic sea ice model (Losch et al., 2010) and combined with a thermodynamic ice model, which is based on the 3-layer model by Winton, (2000) and the energy-conserving LANL CICE model (Bitz and Lipscomb, 1999). The model parameterizes the sea ice/snow albedo as a composite value that integrates contributions from both bare ice and overlying snow. Ice albedo is formulated using an exponential decay function dependent on ice thickness. Snow albedo incorporates both thermal state and aging effects, where fresh snow albedo varies linearly with surface temperature between cold and warm limits, while aged snow albedo decays exponentially with snow age toward an asymptotic old-snow value. The combined surface albedo is then calculated by weighting ice and snow albedos through an exponential

attenuation function based on snow depth (detailed description of the sea ice model can be found in Section 1 of the SI).

The model simulated a period of one year, from May 1 to next April 30, intending to conduct a full melting period followed by a complete freezing phase in the model, which helps to better investigate the effects of SIMW on sea ice melting in summer, as well as its impact on subsequent freezing in winter. Linders and Björk, (2013) used a similar method in their study, but in order to experience a freezing season followed by a thawing season, their experiment was conducted from September to the next August.

**2.2 Initial conditions**

For the ocean initial conditions, the initial profiles of temperature and salinity used in the model runs (Figure 1) are conductivity-temperature-depth (CTD) profiles taken from the World Ocean Database 2023 (WOD23, Mishonov et al., 2024) and Ice-Tethered Profiles (ITP, Krishfield et al., 2008; Toole et al., 2011). We selected 20 profiles as the initial profiles for the 1D model, distributed in different basins of the Arctic Ocean (Figure 1). To capture representative characteristics of each region in the Arctic Ocean, these profiles were grouped geographically into four categories: the Beaufort Sea (BS), the Northern Amerasian Basin (NA), the Amundsen Basin (AM) and the Nansen Basin (NS). The buoyancy frequency profiles show that ocean stratification gradually weakens from the Pacific-influenced sector towards the Atlantic-influenced sector (Figure 2). Within each group, the profiles generally exhibit consistent features, only two individual stations display slight deviations from their group's typical patterns. Specifically, station NA-5, which is located in the NA but close to the central Arctic, exhibits higher salinity and a warmer AWW layer, with its profile more similar to those from the AM. Meanwhile, station NS-1 exhibits characteristics closer to those of the AM region, with notably stronger upper-ocean stratification than stations NS2–NS5. Overall, these 20 profiles demonstrate the diverse stratification characteristics present throughout the Arctic Basin. The chosen profiles are from the late of the freezing season, February to May, between 2011-2023 (color scale in Figure 2), in order to match the beginning of the model simulations, close to the beginning of the melting season.

For the initial conditions of sea ice model, based on satellite observations, between 2011 and 2020, the basin-averaged Arctic sea ice thickness is $1.87 \pm 0.10$ m (mean $\pm$ standard deviation) at the start of the

melting season in May (Landy et al., 2022). Therefore, each set of experiments includes two different initial sea ice thickness (SIT) conditions, 1.5 m and 2 m, to represent the possibly different melting cycles within thin (seasonal) or thick (multi-annual) sea ice scenarios. The initial sea ice concentration (SIC) is about 96%, climatological Arctic basin-averaged, north of the Bering Strait and the Fram Strait and Excluding the Canadian Archipelago, for May 2011-2023 calculated from the sea ice concentration dataset of National Snow and Ice Data Centre (NSIDC). The initial snow thickness of the model is set to 0.19 m, based on the basin-averaged Arctic snow thicknesses in April observed by satellite (Kacimi and Kwok, 2022; Kwok et al., 2020).

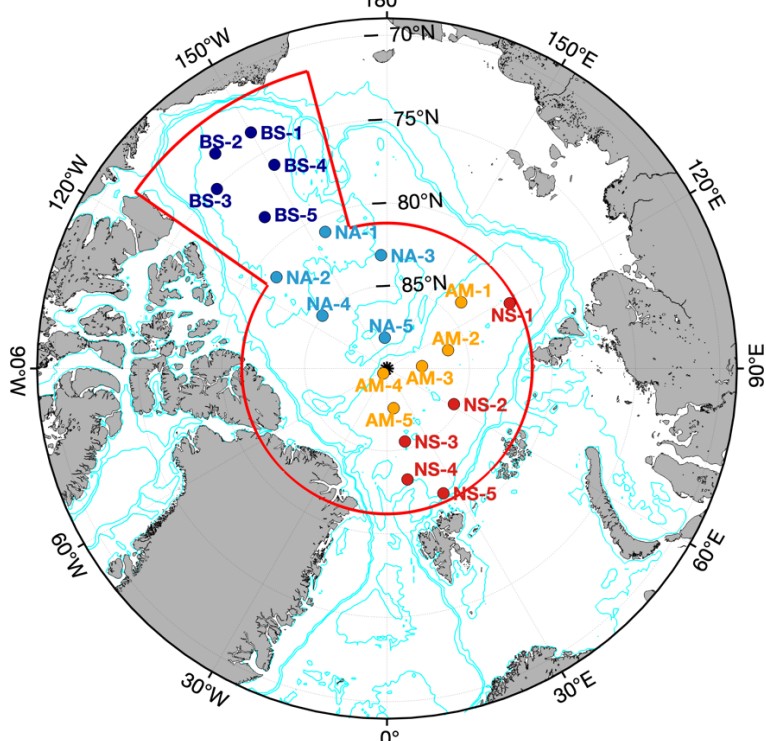

**Figure 1. Locations of the observational data used as initial profiles in the model. The same atmospheric forcing field, based on the 2011-2023 climatological daily average for the region defined by the solid red line, is used in all experiments. BS1 to BS5, located in the Beaufort Sea (dark blue dots); NA1 to NA5 located in the North of the Amerasian Basin (cyan dots); AM1 to AM5 located in the Amundsen Basin (the green dots); NS1 to NS5 located in the Nansen Basin (the red dots). BS: Beaufort Sea; NA: North of the American Basin; AM: Amundsen Basin; NS: Nansen Basin. The thin blue lines represent isobaths of 100, 500, 1000 and 3000 m.**

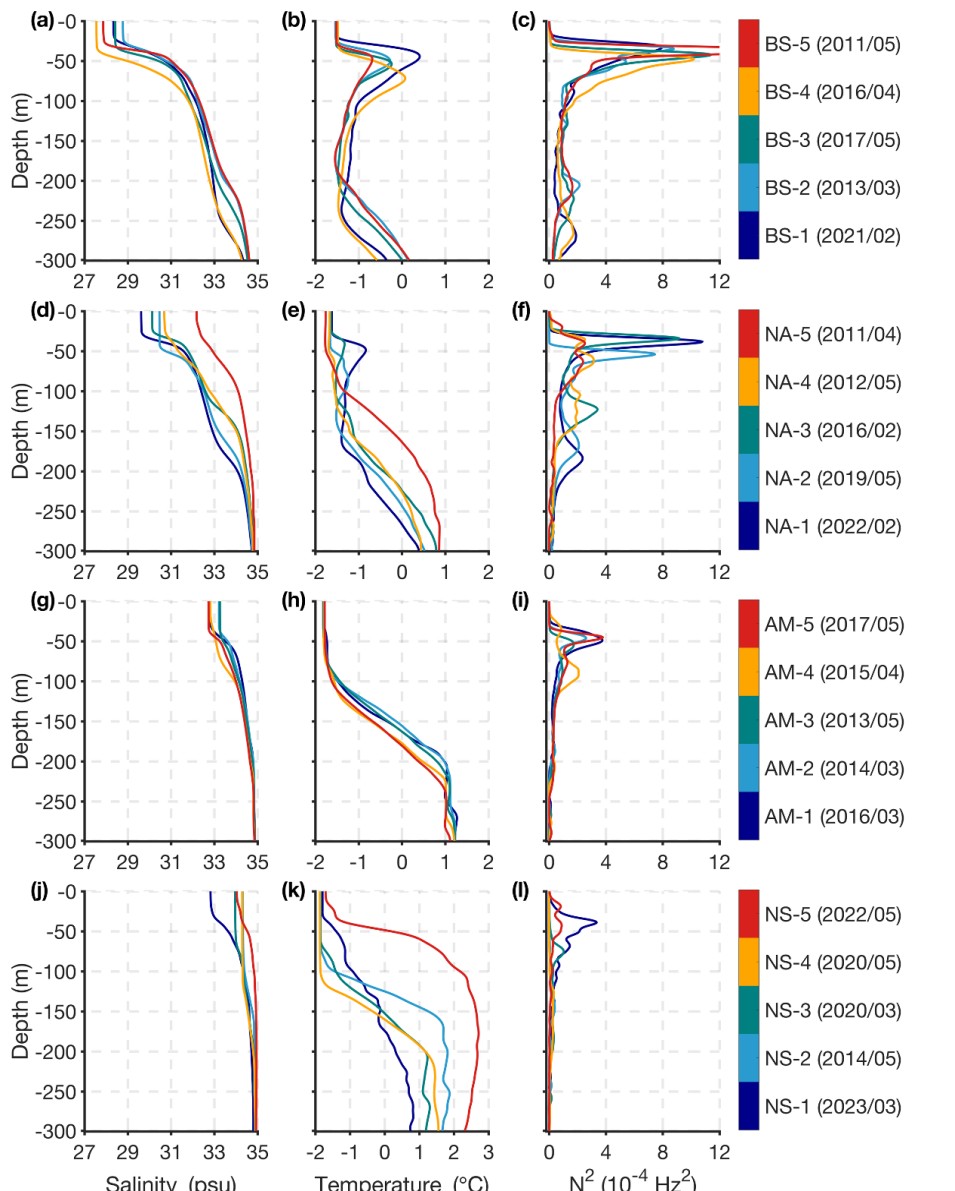

**Figure 2.** Observed salinity (left column), temperature (middle column) and buoyancy frequency (right column) from observational profiles between 2011 and 2023 in the Arctic Ocean, used as initial ocean profiles in the model simulations. To capture representative characteristics of each region in the Arctic Ocean, these profiles were grouped geographically into four categories: (a)-(c): Beaufort Sea (BS); (d)-(f): North of the Amerasian Basin (NA); (g)-(i): Amundsen Basin (AM); (j)-(l): Nansen Basin (NS). The time of each profile is shown in the color scale at the right-hand side. In order to emphasize the regional differences of the upper stratification, we show only the upper 300m. In fact, in the model we used the upper 700 m of data as the initial ocean conditions.

## 2.3 Forcing

The atmospheric forcing for the 1D model is derived from daily averages of atmospheric variables (10 m wind speed, 2 m air temperature, specific humidity, sea level pressure, downward longwave and shortwave radiation) based on the National Centres for Environmental Prediction-Department of Energy (NCEP-DOE) Reanalysis 2 (Kanamitsu et al., 2002). We compute a climatological daily average (2011–2023) by spatially averaging these variables over the entire region enclosed by the red boundary in Figure 1, which uniformly averaged forcing is applied to each experiment. The model also accounts for winter snowfall, as snow cover can insulate the sea ice from the cold atmosphere and thereby regulate winter ice growth (Sledd et al., 2024). Specifically, snowfall in the model begins on October 1st and results in a snow layer approximately 0.19 meters on the sea ice surface by the end of the model period (next April 30th). This value is consistent with both the initial snow condition and the satellite-observed basin-averaged Arctic snow thickness for April (Kacimi and Kwok, 2022; Kwok et al., 2020).

To ensure a more realistic representation of the Arctic freshwater budget and prevent overestimation of SIMW effects, our model includes external freshwater inputs—defined as river runoff, precipitation minus evaporation (P-E), and net inflow/outflow through straits, originating outside the ice-ocean system. For external freshwater inputs, Davis et al., (2016) utilized a time- and area-averaged method in their 1D model. Haine et al., (2015) reported that net inflow of freshwater into the Arctic Ocean is approximately 1200 $km^3$ per year (sum of Runoff, P-E, and the net flow through arctic straits), when it divided by the area of the Arctic ($9.7 \times 10^6\ km^2$), this corresponds to an average input of $3.92 \times 10^{-9}\ m/s$ (based on the 365-day), which is applied to all experiments to represent the external freshwater input, and it is verified that it is a reasonable choice in this 1D model (Figure S5 and S6).

## 2.4 Experiments

Based on the feedback factor (γ) framework proposed by Goosse et al. (2018), when only one feedback is operating, the feedback factor γ can be quantified as the ratio between the additional changes specifically due to the feedback and the response of the full system including all the feedbacks (Total Response, TR). This additional change is itself computed as the difference between the response of the

full system and the one of a reference system in which the feedback under consideration does not operate (Reference Response, RR), if $\gamma$ is a positive (negative) value, it represents positive (negative) feedback:

$$\gamma = \frac{TR-RR}{TR} \qquad (1)$$

In the coupled sea ice-ocean model, the feedback induced by SIMW discharge during melting season is intricately coupled with ice-albedo feedback driven by ice loss, making their independent roles inseparable through conventional single-variable experiments. To overcome this limitation, we designed four progressively constrained sensitivity experiments (Table 1). We artificially create a system devoid of SIMW feedback by setting the SIMW flux to zero (Figure 3b). In addition, we switch off the ice-albedo feedback by maintaining constant summer sea ice concentration and thickness (Figure 3c). Figure 4 shows details of the time development of the simulation results for station BS5 and BS2 with initial SIT of 2 m, helping to understand the procedures of the four experiments more clearly, and the detailed analysis of these results are present in the Section 3.

(1) CTRL: the baseline experiment, which simulates natural ice melting processes with the full coupled ice-ocean system including all the feedback processes (Figure 3a).

(2) noMW: the SIMW flux is set to zero during the melting season in the model to exclude SIMW feedback, while allowing ice thickness to decrease freely (Figure 3b). Figure 4 illustrates a reduction in sea ice during the summer (Figure 4b and c), while the SIMW flux remains zero (Figure 4d).

(3) noIA: we artificially maintain constant sea ice thickness and concentration during the melting season in the model (i.e., fixed to the value of the first melt-day) to keep the ice-albedo constant, while still permitting SIMW discharge. As shown in Figure 4, summer sea ice thickness, concentration and ice-albedo remain unchanged (Figure 4e and f) during the melting season, but SIMW flux is not zero (Figure 4g and h).

In addition, we also want to explore the net ice-albedo feedback and SIMW feedback. Therefore, we designed the following experiment:

(4) noMWIA: based on the noIA configuration but setting SIMW flux to zero (Figure 3d) to eliminate both ice-albedo feedback and SIMW feedback. As shown in Figure 4, during the melting season, summer sea ice thickness, concentration and ice-albedo remain unchanged (Figure 4a-f), and SIMW flux is zero (Figure 4g-h).

There are two points to note: First, in the noIA and noMWIA run, we artificially held sea ice
constant in the model to limit the ice-albedo feedback, which means we cannot get the actual ice melting
thickness from the sea ice diagnostic of the model output. However, we can obtain it as an offline
diagnostic using Eq. (S15) as described in the SI. Second, all imposed constraints are enforced exclusively
during the melting season, with the model reverting to natural ice-ocean coupling during freezing season.
This design enables explicit assessment of how summer-imposed interventions modulate the winter
evolution of the naturally coupled ice-ocean system.

These experiments systematically decouple the effects of SIMW discharge from ice-albedo
feedback caused by ice loss through imposing targeted variable constraints, enabling to quantify precisely
both the independent SIMW feedback and its relative contribution compared to ice-albedo feedback.
Through these four experimental sets, combined with the methodology of Goosse et al., (2018), we
quantitatively assess the feedback factors (detailed in Section 4). Although our experimental design is
somewhat idealized, this approach can directly illustrate the independent impacts of SIMW on the ice-
ocean system and their relative importance. We evaluated this 1D model against observations in several
aspects, including the seasonal variation of vertical temperature-salinity structure, the volume of
meltwater release, ice-ocean heat flux, and ice-albedo values. The results demonstrate that this simplified
model can replicate the observed seasonal cycles of these key physical variables in the ice-ocean system
well (see Section 2.4 in the SI).

**Table 1. Experiment matrix**

| Experiment | Sea ice evolution | SIMW Flux | Purpose |
|---|---|---|---|
| (1) CTRL | Prognostic | Natural | Baseline under natural state |
| (2) noMW | Prognostic | Zeroed | No SIMW feedback |
| (3) noIA | Fixed (first melt-day state) | Natural | No ice-albedo feedback |
| (4) noMWIA | Fixed (first melt-day state) | Zeroed | No SIMW and ice-albedo feedbacks |

**Note: all imposed constraints are enforced exclusively during the melting season, with the model reverting to natural ice-ocean**
**coupling during freezing season.**

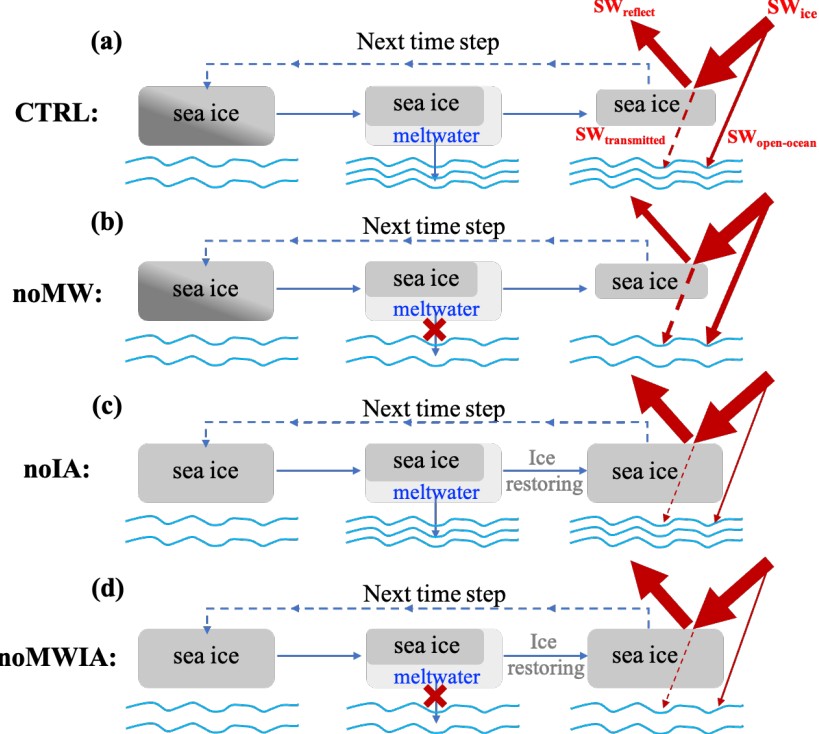

**Figure 3. Schematic representation of the four experiments in this study. (a): CTRL (Control experiment), with ice-ocean coupled model running normally; (b): noMW (no SIMW feedback experiment), after sea ice melting, the SIMW flux entering the ocean is set to 0; (c): noIA (no ice-albedo feedback experiment), after sea ice melting, SIMW enters the ocean normally, but before the model starts the next time step, sea ice is restored to its first melt-day state; (d): noMWIA (no ice-albedo feedback and SIMW feedback experiment), after sea ice melts, the SIMW flux entering the ocean is set to 0, and simultaneously, before the model starts the next time step, the sea ice is restored to its first melt-day state. The red arrows represent the shortwave radiation fluxes: SWice (Shortwave radiation reaching the ice surface), SWreflect (shortwave radiation reflected from the ice surface), SWtransmitted (shortwave radiation transmitted through sea ice to the ocean), and SWopen-ocean (shortwave radiation entering the ocean through open ocean).**

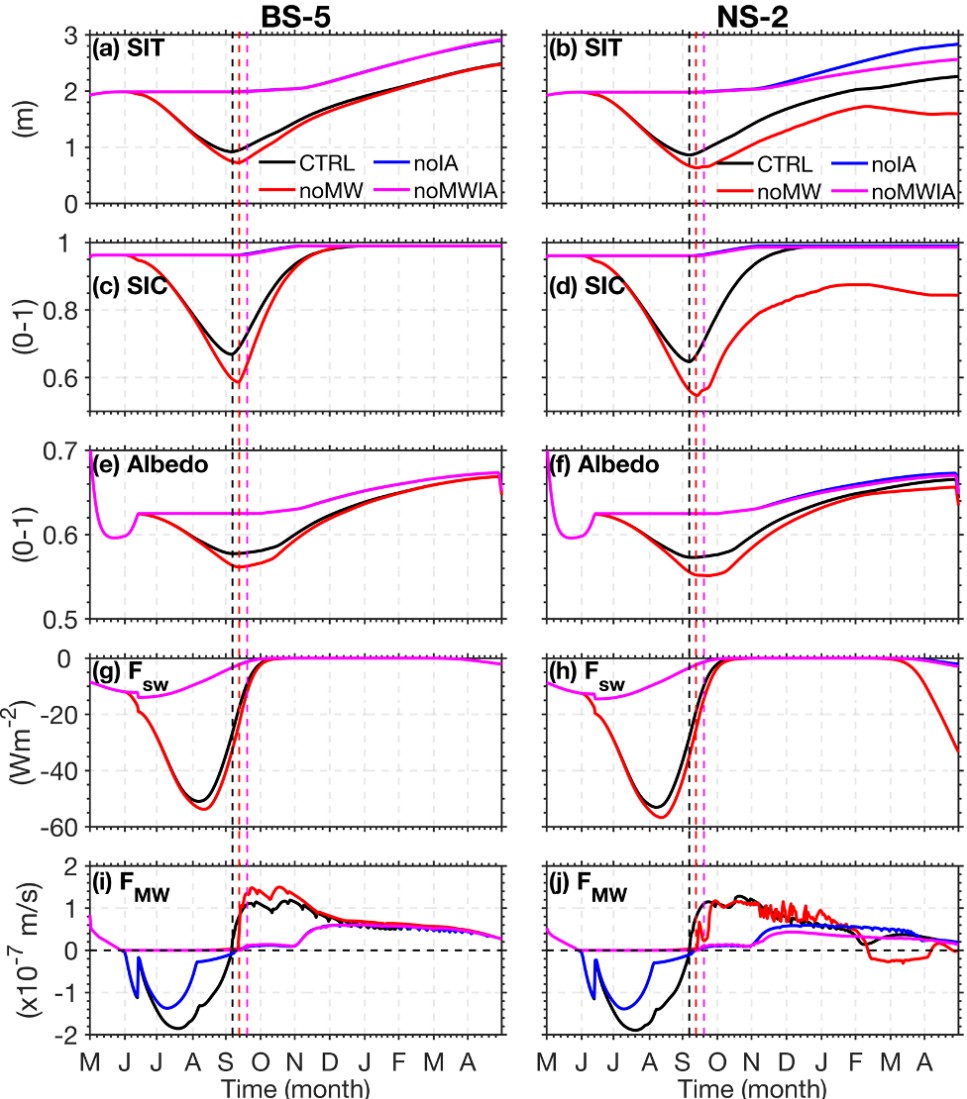

Figure 4. Modeled (a)-(b) sea ice thickness (SIT); (c)-(d) sea ice concentration (SIC); (e)-(f) ice/snow albedo; (g)-(f) net ocean
shortwave heat flux ($F_{SW}$), negative values represent heat entering the ocean. (i)-(j) SIMW flux ($F_{MW}$), negative values represent
freshwater entering the ocean. The left column is the result of station BS-5 and the right column is the result of station NS-2. The
dashed lines perpendicular to the X-axis represent the first freezing day of each experiment (defined as the first day on which the
ice thickness growth rate surpasses 0.1 cm/day).

## 3 Model results

The model results are not sensitive to small changes in initial ocean profiles. Therefore, the experimental results for most of the initial profiles show qualitative consistency, except for the weakly stratified profiles, i.e., NS2-NS5, located in the Nansen basin. In order to show the general model behavior and the impact of ocean initial profiles, the detailed time developments of some model diagnostics are shown for two basic types of initial profiles, stations BS5 (strong stratification) and NS2 (weak stratification, as shown in Figure 2). For the other stations, we provide their modeled diagnostic averaged values in the melting and freezing season. In each experiment, the period before the first freezing day in September is defined as the melting season, while the period from the first freezing day to the end of the simulation is the freezing season. The first freezing day is defined as the first day on which the ice thickness growth rate exceeds 0.1 cm/day. This section primarily presents and discusses the results from experiments with initial SIT of 2 m. Results for experiments with initial SIT of 1.5 m are provided in the SI.

### 3.1 Ocean Response

#### 3.1.1 Melting season

In the CTRL run, SIMW release during summer forms a low-salinity layer, establishing a summer shallow mixed layer of 10 meters (Figure 5e and Figure 6a). This layer traps and stores a portion of solar heat beneath the shallow mixed layer, leading to the development of a warm water mass known as the NSTM below the mixed layer (Figure 5a). The NSTM forms because the low-salinity SIMW layer strengthens the vertical density gradient, limiting convective heat loss and allowing it to persist as a residual warm layer at the base of the mixed layer (Alvarez, 2023; Jackson et al., 2010; Steele et al., 2011). In our simulation, the NSTM layer is typically observed at depths of approximately 10–25 m (Figure 5a), consistent with observed NSTM depths (Gallaher et al., 2017; Jackson et al., 2010). Additional information on model validation can be found in Section 2.4 and Figure S7 in the SI.

However, when SIMW feedback is disabled (noMW run), this vertical structure fundamentally changes. During the melting season, the removal of SIMW results in to a deeper, colder, and saltier upper

ocean (Figure 5). Particularly at stations with weak initial stratification (e.g., NS2-NS5), SIMW removal causes a significant deepening of the MLD during summer (Figure 5j and n). A prominent result is the disappearance of the NSTM in the noMW and noMWIA run (Figure 5b and d). For example, at station BS5 (strongly stratified), SIMW removal increases the summer MLD to approximately 30 m (Figure 5b), which prevents NSTM formation because, in the absence of SIMW-induced summer shallow mixed layer, the heat absorbed by the ocean mixes thoroughly within the upper 30 m, resulting in a uniform temperature profile (Figure 5b), rather than a warm layer isolated at the mixed layer base (Figure 5a). This results in the heat that could originally be stored in the NSTM at the bottom of the summer shallow mixed layer directly contacting the ice bottom, causing more sea ice melting, which will be discussed in detail in Section 3.2.

Completely insulating the ice-albedo feedback (noIA run) leads to the ocean absorbing less solar heat (Figure 4g and h, blue and magenta lines) and results in the NSTM being cooler in the noIA run than in the CTRL run (Figure 5a and c). When both the SIMW feedback and the ice-albedo feedback are removed (noMWIA run), the mixed layer becomes deeper and colder compared to when only the ice-albedo feedback is removed (Figure 5c and d). It should be noted that in the CTRL and noIA run, the MLD shows a discontinuity during the early melting season. This occurs due to the input of SIMW into the ocean. Specifically, when sea ice begins to melt, the buoyancy flux term changes sign. This change leads to the formation of a new, shallower, and fresher surface layer. As a result, the MLD shoals rapidly. In contrast, the noMW and noMWIA run do not exhibit this discontinuity. This is because SIMW is removed in these two experiments.

### 3.1.2 Freezing season

During the freezing season, in the CTRL run, with the freezing and salt rejection process, the mixed layer gradually deepens, and the depth of the NSTM also gradually increases, with the temperature gradually decreasing until it disappears (e.g., Figure 5a and i). In the noMW run, a notable phenomenon is that the stations NS2-NS5 located in the Nansen basin exhibit strong vertical mixing and much deeper MLD than other stations (Figure 6b). For example, at station NS2, the MLD deepens rapidly after December and reaches the core depth (~300 m) of the AWW by February, resulting in the upward mixing

of AWW, which leads to warming (and salinification) of the upper layer and cooling (and freshening) of the AWW layer (Figure 5r and v). The pronounced impact of SIMW on MLD during the freezing season at stations NS2–NS5 can be attributed to the interplay between pre-existing weak stratification and the absence of SIMW-induced freshwater input. These stations inherently exhibit weak stratification (Figure 2l). In the noMW run, the removal of summer SIMW results in higher surface salinity and further weakens stratification compared to the CTRL run (Figure 5u and v). As winter begins, sea ice formation drives brine rejection and associated convection. In the noMW run, the lack of residual SIMW leads to stronger brine rejection convection, which eventually overwhelms the already weak stratification, leading to intense vertical mixing and a rapid deepening of the MLD. In contrast, in the CTRL run, the presence of surface SIMW maintains sufficient stratification to resist the convection induced by freezing, thereby limiting mixed layer deepening. Observations have shown that in the Nansen basin with weak stratification, winter brine rejection can drive intense vertical mixing and upward heat flux from the AWW layer, significantly impacting sea-ice growth (Polyakov et al., 2017, 2020).

However, at stations in the Canadian and Amundsen basins, due to stronger initial stratification, while the mixed layer obviously deepened in winter after removing the SIMW (Figure 6b), it still could not reach the AWW in the deep (e.g., Figure 5b and 5j). For example, at station BS5 with stronger stratification, even with SIMW removed (no-MW run), the maximum MLD in winter increased only from 31 meters in the CTRL run to 37 meters, which cannot reach the depth of the Pacific summer warm water layer (Figure 5b). At station AM2, which has intermediate stratification weaker than BS5 but stronger than NS2, removing SIMW increased the maximum MLD from 49 meters in the CTRL run to 92 meters in the noMW run, deepening by approximately 40 meters. Although the vertical mixing is more pronounced compared to station BS5, it still cannot reach the core depth of the AWW layer (Figure 5i).

In the noIA and noMWIA run, where the ice-albedo feedback is removed (by remaining summer sea ice constant), the deepening of the winter MLD is smaller. At station AM2, the MLD in the noMWIA run only deepened by about 10 meters compared to the noIA run (Figure 5k and l), much smaller than the 40-meter difference between the noMW run and the CTRL run. In the noMWIA run, the MLD during winter at station NS2 is only slightly deeper than in the noIA run (Figure 5s and t). Although the noMWIA run also removed SIMW, it does not exhibit the strong vertical mixing observed in the noMW run,

indicating that the presence of sea ice effectively inhibits ocean vertical mixing and mitigates the effects

of removing SIMW.

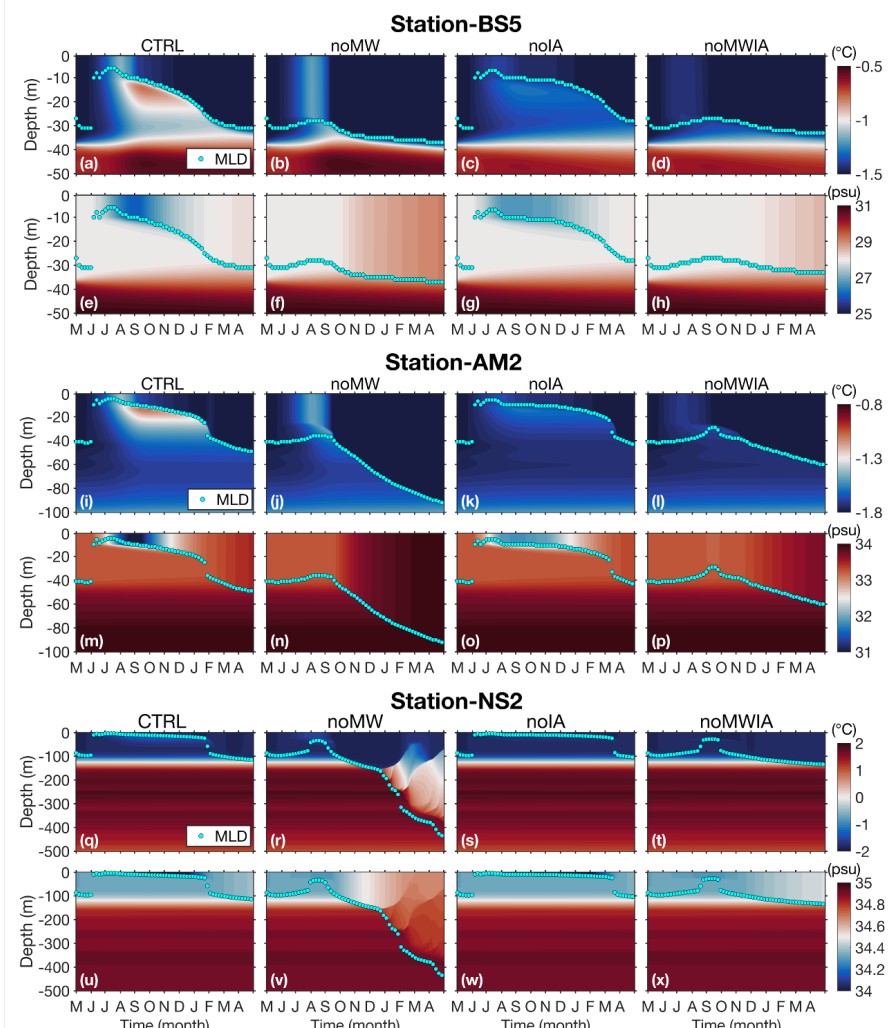

**Figure 5. Simulated vertical profiles of ocean temperature and salinity over time at stations (a)-(h) BS5, (i)-(p) AM2, and (q)-(x) NS2. For each station, the top row displays temperature, and the bottom row displays salinity. From left to right, the columns correspond to the CTRL run, noMW run, noIA run, and noIAMW run. Cyan dots in each panel represent the mixed layer depth. Note that the vertical depth scales are not consistent across stations.**

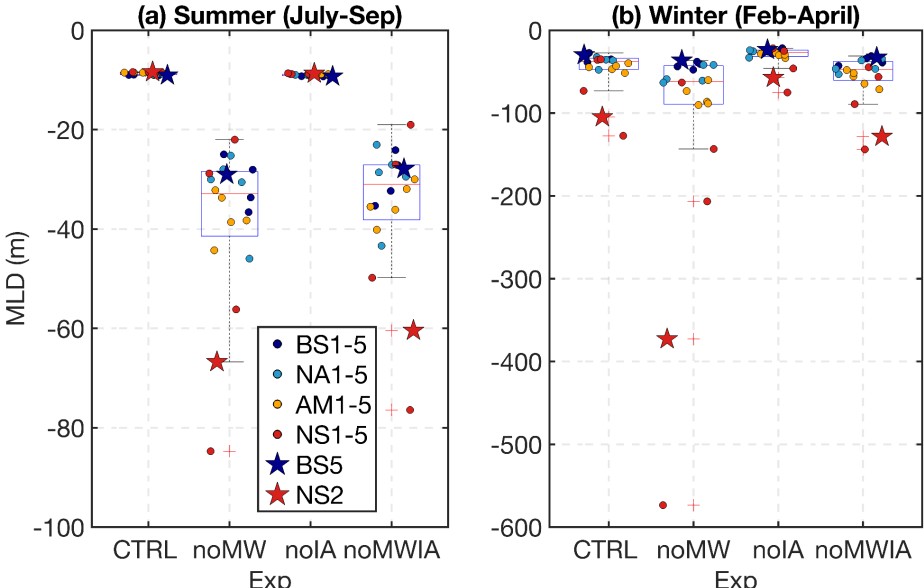

**Figure 6. Box plots illustrating the mean (a) MLD in summer and (b) winter across different Stations in different types of experiments. Each box plot shows the median, interquartile range, and potential outliers (points marked with red plus sign). All points are the results of experiments with initial SIT of 2 m.**

## 3.2 Sea Ice Response

### 3.2.1 Melting season

Figure 7a shows the thickness of sea ice melted during summer. When the SIMW feedback is disabled (noMW run), summer ice melting increases by $0.21 \pm 0.01$ m compared to the CTRL run, averaged value of experiments with initial SIT of 2 m. This implies that SIMW inhibits ice melting, thus establishing a negative feedback mechanism. This is because the absence of the stabilizing surface SIMW layer enhances vertical mixing (as shown in Fig. 5b), allowing heat stored in the NSTM, which would otherwise be isolated from the sea ice by the SIMW layer, to directly heat the ice base, thereby increasing summer sea ice melt. As demonstrated in Figure 8a and b, the noMW run exhibits higher ocean-to-ice heat flux than CTRL run, leading to an additional ice melting. The ocean-to-ice heat flux has a significant seasonal cycle with maximum values reaching 40-60 Wm$^{-2}$ in summer (Maykut and McPhee, 1995) and close to zero during winter in many instances (Krishfield and Perovich, 2005; McPhee et al., 2003; Zhong

et al., 2022). The model results of the CTRL run are in agreement with the observed values well (e.g.,
Fig. 8a and b, black lines).

Disabling ice-albedo feedback (noIA run) reduces ice melting by $0.45 \pm 0.02$ m compared to the
CTRL run (Figure 7a), demonstrating that ice-albedo feedback is positive feedback that enhances summer
ice melting. This is because disabling the ice-albedo feedback (noIA run) maintains a constant summer
albedo (Figure 4e, blue line), which significantly reduces the shortwave radiation entering the ocean
compared to the CTRL run (Figure 4g, blue vs. black line). This decrease in absorbed radiation
subsequently lowers the ocean-to-ice heat flux (Figure 8a, blue vs. black line), ultimately reducing ice
melting.

When both feedbacks are suppressed (noMWIA run), sea ice melting decreases by $0.39 \pm 0.02$ m
relative to the CTRL run (Figure 7a). It is also evident that under condition isolated from ice-albedo
feedback (noIA and noMWIA run), the presence of SIMW feedback (noIA run) resulted in only 0.06 m
less ice melting than its absence (noMWIA run), demonstrating the negligible impact of SIMW feedback
without ice-albedo feedback run (Figure 7a). This is because when ice-albedo feedback is disabled, two
interrelated effects arise: first, the reduction in sea ice melting diminishes SIMW release, thereby
weakening the stratification-enhancing effect of SIMW, as indicated by the notably fresher surface water
in the CTRL run compared to the noIA run (Figs. 5e and 5g). Second, the decreased shortwave radiation
entering the ocean reduces the heat source available for isolation in the NSTM by the SIMW layer, evident
from the higher NSTM temperatures in the CTRL run relative to the noIA run (Figs. 5a and 5c).
Collectively, these effects substantially attenuate the efficacy of the SIMW feedback in the absence of
ice-albedo feedback. Furthermore, under conditions isolated from SIMW feedback (noMW and noMWIA
run), the presence of ice-albedo feedback increased melting by $0.60 \pm 0.02$ m compared to its absence
(Figure 7a). These findings show that the ice-albedo feedback is the dominant process governing the
Arctic sea ice-ocean system, which also significantly amplifies the efficacy of the SIMW-induced
negative feedback.

### 3.2.2 Freezing season

During the freezing season, sea ice evolution can be influenced by the stratification of the initial ocean profile (Figure 7b). In the CTRL run, stations NS2-NS5 with weak stratification exhibit less ice formation in winter compared to strongly stratified regions (Figure 7b, CTRL). This difference is governed by how ocean stratification modulates vertical heat transport. In strongly stratified regions, a pronounced halocline acts as a stable barrier. This halocline is primarily formed and sustained by a ventilation process: intense freezing in seasonal ice zones over the shelf seas generates cold, dense, and saline waters, which flow into the deep basins along isopycnal surfaces and feed the halocline (Itkin et al., 2015; Metzner et al., 2020). Because of this pre-existing, stable layer with a strong salinity gradient (e.g., Figure 2a), local wintertime brine rejection is insufficient to erode it, making it difficult to bring deep heat to the surface layer, thereby promoting ice formation. Conversely, at stations NS2-NS5 where stratification is weak, the halocline is shallow and unstable, and the warm AW is closer to the surface (e.g., Figure 2j and k). Here, vertical convection induced by local brine rejection readily entrains heat from the underlying AW. The resultant increase in the ocean-to-ice heat flux thereby suppresses sea ice formation. This mechanism is consistent with Linders and Björk, (2013), who demonstrated that weak salinity stratification allows winter ocean heat fluxes of up to 8 $W/m^2$ to reach the ice, whereas a well-developed halocline reduces fluxes to ~0.7 $W/m^2$. Similarly, Polyakov et al., (2013) observed that winter convection in the eastern Arctic Ocean transports heat from AWW to the surface layer, reducing ice formation in weakly stratified areas.

Removing SIMW (noMW run) further amplifies these contrasting responses caused by the initial ocean stratification difference. In the noMW run, ice formation in the strongly stratified regions is greater than in the CTRL run (Figure 7b). For instance, at station BS5 with strong stratification, the removal of SIMW (noMW run) leads to a 0.19 m increase in ice formation compared to the CTRL run (Figure 7b). This is because, enhanced summer ice melt in the noMW run results in thinner sea ice and more extensive open water by the start of winter (Figure 4a and c). These conditions favor rapid ice formation in early winter through two key mechanisms. First, the thinner ice allows more efficient thermal conduction, speeding up ice basal cooling during early winter (Figure 8c). Second, the larger areas of open-ocean promote stronger heat loss from the ocean to the atmosphere, as seen in the elevated ocean-to-atmosphere

heat fluxes in the noMW run (Figure 8e). The synergy of these processes drives faster ice growth, demonstrating that summer ice retreat can paradoxically enhance early winter ice formation by improving both conductive and surface heat loss. This phenomenon is consistent with findings from previous studies. Analyses of CMIP6 model data and ice mass balance buoys indicate that basal growth increases in winter due to thinner sea ice during the 21st century (Keen et al., 2021; Lin et al., 2022).

In contrast, in the weakly stratified regions, the precondition of weak stratification allows the removal of SIMW to trigger a dramatic regime shift, ultimately leading to a net reduction in winter ice formation (noMW in Figure 7b). For instance, at station NS2, winter ice formation is reduced by 0.62 m in the noMW run. Although early winter ice formation is initially faster, the weak stratification is unable to resist the convection driven by brine rejection. This leads to the intense upward mixing of AWW, as shown in Figure 5r. The consequence of this deep heat reaching the surface is a dramatic surge in the ocean-to-ice heat flux during late winter (Figure 8b, red line), which peaks at ~30 W/m$^2$—a value high enough to induce basal ice melt in winter. This phenomenon is more pronounced in experiments with the initial SIT of 1.5 m (Figure S8-S10 in the SI). In addition, we also used climatological data from World Ocean Atlas 2023 (WOA2023, Reagan et al., 2024) as the ocean initial conditions to conduct simulations in each basin (Figure S11 in the SI), and the results persistently show that the removal of SIMW leads to upward mixing of AWW and ice melting in winter in western Nansen Basin (Figure S12-S14 in the SI). These results based on the climatological conditions demonstrate that the intense vertical mixing and ice melting during the winter in the Nansen Basin is not a coincidental event caused by the initial profile selection.

The influence of SIMW is further demonstrated by comparing the noMWIA and noIA experiments. As shown in Figure 5b, the noMWIA run exhibits reduced sea ice formation in the NS2, NS4, and NS5 regions relative to the noIA run. In the absence of SIMW, weakened stratification intensifies winter convection and upward heat transport, increasing the ocean-to-ice heat flux (Figure 8b) and thereby slowing ice growth in these regions (Figure 4b). However, unlike the noMW run where strong winter vertical mixing and sea ice melt are observed (Figure 4b and Figure 5r), these phenomena do not occur in the noMWIA run. This discrepancy arises because the noMWIA experiment maintains summer sea ice cover unchanged, leading to thicker sea ice and a much smaller open ocean by the onset of winter.

Consequently, both ice basal and oceanic heat loss proceed slowly (Figure 8c–f), reducing the winter ice

formation. The associated weakening of brine rejection further weakens the ocean vertical convection,

which becomes insufficient to disrupt the pre-existing stratification. As a result, AWW cannot reach the

surface (Figure 5t), and winter ice melt does not occur (Figure 4b) in the noMWIA run.

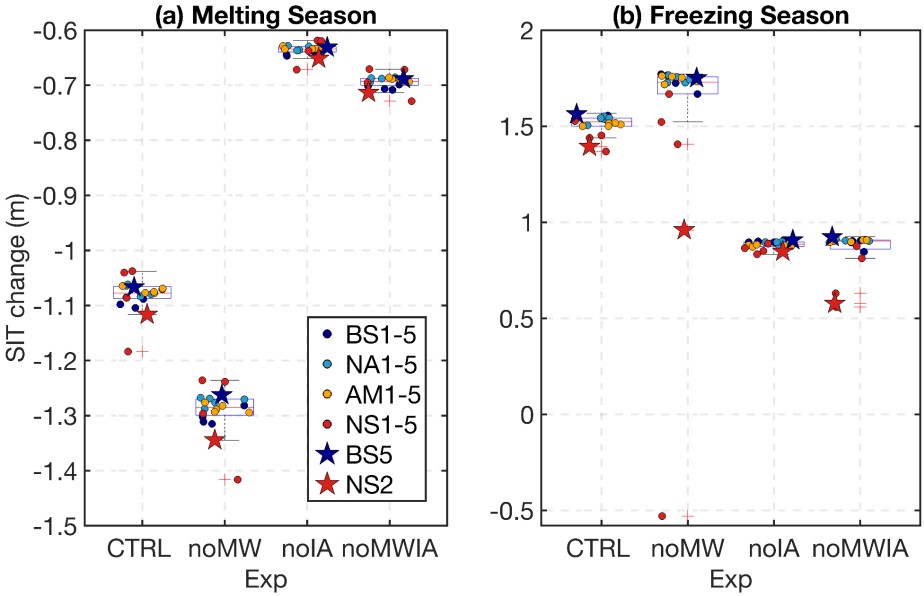

**Figure 7. Box plots illustrating the (a) ice thickness changes during the melting season and (b) freezing season across different**

**Stations in different types of experiments. Each box plot shows the median, interquartile range, and potential outliers (points marked**

**with red plus sign). All points are the results of experiments with initial SIT of 2 m.**

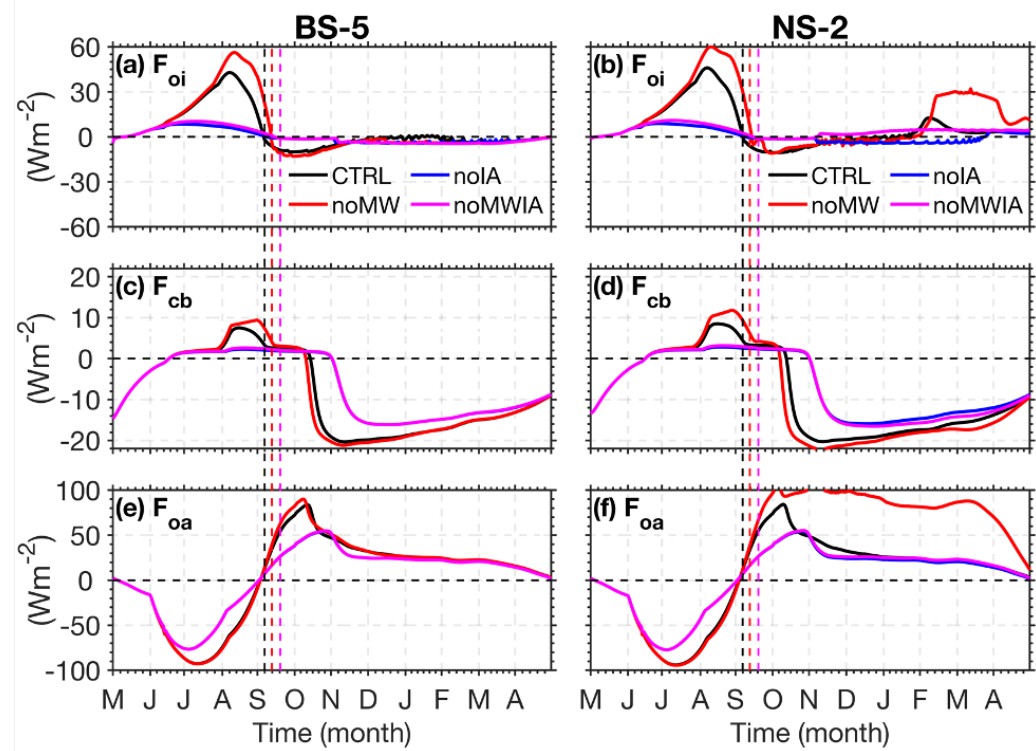

**Figure 8. Heat flux time series at station BS5 (left column) and NS2 (right column). (a)-(b):** $F_{oi}$ **(ocean-to-ice heat flux); (c)-(d):** $F_{cb}$

**(heat flux caused by thermal conduction from the ice surface to base); (e)-(f):** $F_{oa}$ **(ocean-atmosphere heat flux over the open ocean)**

**for the experiments with initial SIT of 2 m. In the panels of (a)-(d), positive (negative) values denote heat gain (loss) at the ice base.**

**In the panels of (e)-(f), positive (negative) values denote upward (downward) heat flux, corresponding to oceanic heat loss (gain).**

## 4 Quantifying Feedback Factors

This section quantifies SIMW and ice-albedo feedback using the feedback factor (γ) framework

proposed by Goosse et al. (2018). Here, we focus on the SIMW and ice-albedo feedbacks to ice melting

during the melting season, so, we selected the thickness of sea ice melted during the melting season as

the reference variable. Based on the details of these decoupling experiments provided in Table1 and

Figure 3, we can calculate the ice-albedo feedback factor ($\gamma_{IA}$) and SIMW feedback factor ($\gamma_{MW}$).

$\gamma_{IA} = \frac{TR_{CTRL} - TR_{noIA}}{TR_{CTRL}}$           (2)

$\gamma_{MW} = \frac{TR_{CTRL} - TR_{noMW}}{TR_{CTRL}}$          (3)

here, $TR_{CTRL}$, $TR_{noIA}$ and $TR_{noMW}$ are the thickness of sea ice melted during the melting season in the CTRL (full system including all the feedbacks), noIA (ice-albedo feedback not operating) and noMW (SIMW feedback not operating) run, respectively (as shown in Figure 5a).

Additionally, our experimental design provides a framework to quantify the net effects of SIMW and ice-albedo feedbacks by preventing each from the influence of one another. The net ice-albedo feedback factor ($\gamma_{nIA}$) and the net SIMW feedback factor ($\gamma_{nMW}$):

$$\gamma_{nIA} = \frac{TR_{noMW} - TR_{noMWIA}}{TR_{noMW}} \quad\quad (4)$$

$$\gamma_{nMW} = \frac{TR_{noIA} - TR_{noMWIA}}{TR_{noIA}} \qu\quad (5)$$

where $TR_{noMWIA}$ is the thickness of sea ice melted during the melting season in the noMWIA run (both SIMW and ice-albedo feedbacks not operating).

Figure 9 shows the feedback factors for all stations. We can notice that although Figure 5a shows differences in summer sea ice melt thickness among the stations, the calculated feedback factors remain very close. The results indicate that the positive ice-albedo feedback ($\gamma_{IA} = +0.41$) is approximately twice as strong as the negative SIMW feedback ($\gamma_{MW} = -0.19$). When ice-albedo feedback is fully eliminated, the negative SIMW feedback strength decreases to $\gamma_{nMW} = -0.09$ (Figure 9). The ice-albedo feedback factor only increased from +0.41 to +0.46 after preventing the SIMW feedback. In fact, when the feedback factor is expressed as a percentage, it represents the percentage increase or decrease in sea ice melting relative to the reference experiment (i.e., the denominator in Eq. (2)-(5)). Based on this, a schematic diagram is presented herein, as presented in Figure 10 showing the values of each feedback factor and the percentage change in sea ice melting thickness between experiments.

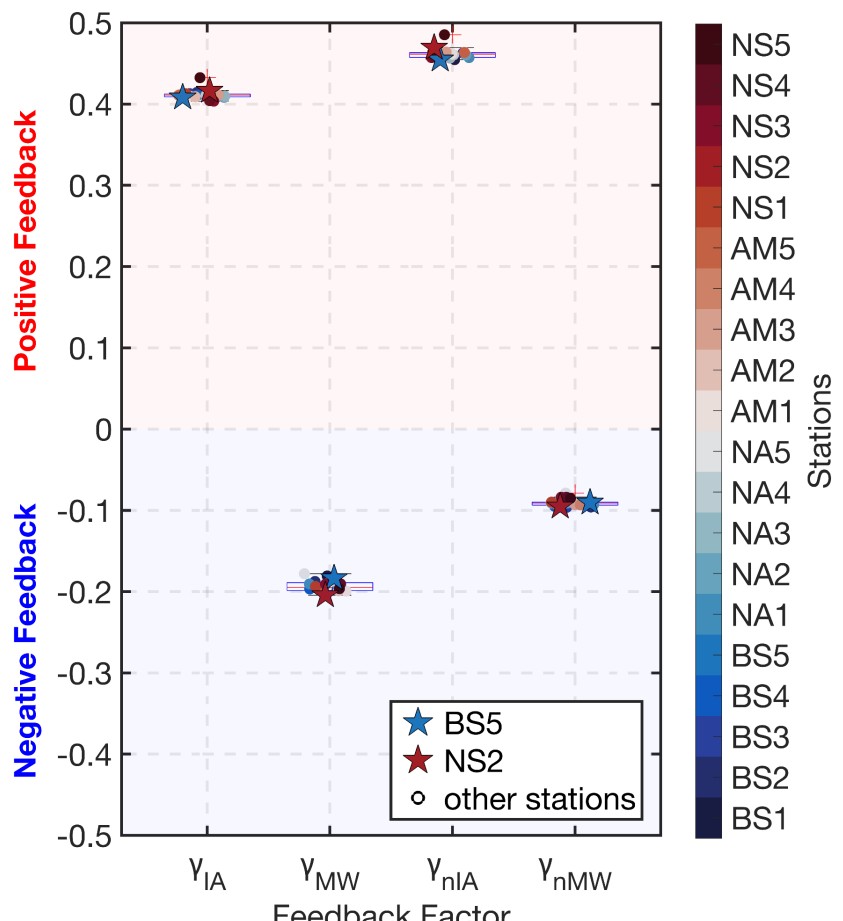

**Figure 9.** Box plots illustrate the four feedback factors across different stations. If the $\gamma$ is a positive (negative) value, it represents positive (negative) feedback. All points are the results of experiments with initial SIT of 2 m. The blue star and red star represent stations BS5 and NS2, respectively. Ice-albedo feedback Factor: $\gamma_{IA}$; SIMW feedback Factor: $\gamma_{MW}$; Net SIMW feedback factor: $\gamma_{noIA}$; Net ice-albedo feedback factor: $\gamma_{noMW}$.

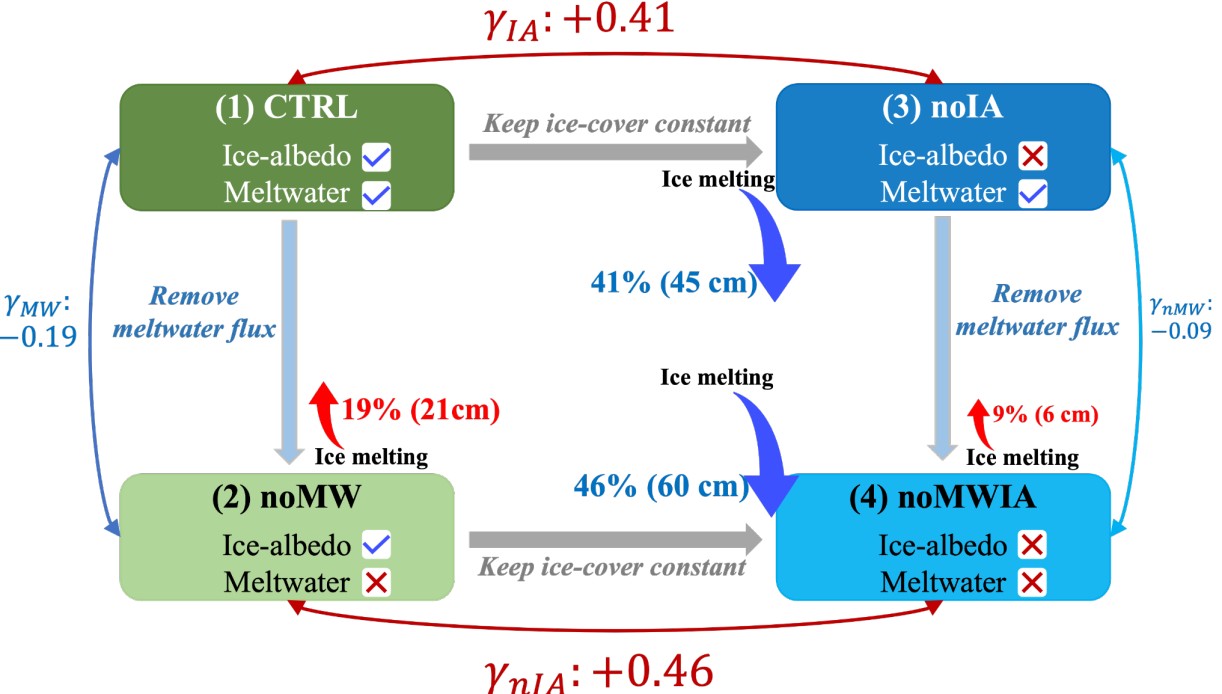

**Figure 10. Schematic of the relationships between four experimental setups for studying ice-albedo and SIMW feedbacks. The blue**
**checkmark indicates that the feedback is existent, and the red cross indicates that the feedback is eliminated. The blue or red double-**
**headed arrow between two experiments signifies the comparison of these results and the corresponding feedback factors obtained.**
**The red upward arrow indicates the increase in ice melting, while the red downward arrow indicates the decrease in ice melting.**
**Left Side: With ice-albedo feedback present, removing SIMW feedback results in a 19% increase in ice melting; Right Side: Without**
**ice-albedo feedback, removing SIMW feedback leads to a 9% increase in ice melting; Top Side: With SIMW feedback present,**
**removing ice-albedo feedback causes a 41% decrease in ice melting; Bottom Side: Without SIMW feedback, removing ice-albedo**
**feedback results in a 46% decrease in ice melting.**

## 5 Discussion

This study focuses on ocean vertical stratification changes influenced by the sea ice melt/freeze
cycle and the subsequent feedback on sea ice. For this purpose, we used a 1D coupled sea ice–ocean
model that excludes advective processes. Although simplified, model validation demonstrates that our 1D
model reproduces the seasonal evolution of Arctic sea ice and upper-ocean hydrographic properties—
such as mixed layer depth and the position and depth of the summer NSTM—in a manner comparable to
observations (Figure S7). On short timescales, 1D models are commonly employed to investigate ocean

vertical stratification changes and ice-albedo feedback (Himmich et al., 2024; Linders and Björk, 2013; Toole et al., 2010; Wang et al., 2024).

Nevertheless, the influence of advection in the Arctic ice-ocean system cannot be overlooked. This is particularly evident in the western Eurasian Basin, where AWW inflow and its associated heat transport significantly impacts the local sea ice-ocean system (Polyakov et al., 2017). The West Spitsbergen Current delivers a heat flux of up to 200 W/m$^2$ into the Arctic Ocean (Aagaard et al., 1987), which then spreads throughout the Arctic Ocean via a cyclonic boundary current and thermohaline intrusion, forming the AWW Layer (Long et al., 2024). In our noMW simulation, even without this advective heat supply, substantial ocean-to-ice heat flux and winter sea ice melt were observed in the Nansen Basin case (such as Figure 8b). This indicates that the winter heat flux from upwelled AWW would be even larger when advective heat input is considered.

Moreover, advective transport of sea ice and surface freshwater driven by atmospheric wind patterns can directly influence freshwater sources across different basins of the Arctic Ocean. For example, during the positive Arctic Dipole (AD) phase, freshwater content decreases in the Eurasian Basin and increases in the Amerasian Basin due to the changes of wind-driven Ekman convergence (Wang, 2021). Under the negative Arctic Oscillation (AO) phase, sea ice in the western and central Eurasian Basin exhibits an anticyclonic drift anomaly that accelerates export, while the positive phase reverses this pattern (Wang et al., 2021b). Although our model does not include advective processes, the results effectively reflect their potential impacts. For instance, the noMW run simulates a scenario similar to the net loss of freshwater. In this run, SIMW removal triggers intense vertical mixing in the Nansen Basin (Figure 5r or Figure S11). This implies that in reality, a positive AD phase could produce comparable effects in these regions. Similarly, a positive AO anomaly, which imports sea ice, could enhance regional stratification stability, analogous to the conditions in our noIA run (with sea ice held constant during summer).

The mechanism by which regional advection-driven sea ice changes affect stratification has been documented in the Barents Sea. Since the mid-2000s, the intense upward mixing of AWW in the Barents Sea has been primarily driven by reduced sea ice inflow (Lind et al., 2018; Skagseth et al., 2020). Sea ice serves as a key freshwater source in this region, maintaining the surface freshwater layer and stable

stratification. A reduction in sea ice inflow directly weakens stratification and enhances vertical mixing (Skagseth et al., 2020). This mechanism is also analogous to the findings from our noMW run in the Nansen Basin, which indicates that, even under current Eurasian Basin conditions, sea ice cover and SIMW release are critical for maintaining stratification and suppressing the upward heat flux from AWW. Many studies have demonstrated that Atlantification is intensifying in the Eurasian Basin (Barton et al., 2018; Muilwijk et al., 2023; Polyakov et al., 2017; Tesi et al., 2021), leading to water properties that increasingly resemble those of the Barents Sea, characterized by warmer temperatures, higher salinity, and weaker stratification. Consequently, as Atlantification advances and Arctic sea ice declines rapidly in the future, the role of SIMW in maintaining stratification will become increasingly important. If sea ice retreats beyond a critical threshold where SIMW production becomes insufficient to sustain stratification and the surface ice cover is too thin to buffer atmosphere-ocean interactions, the basin may experience frequent, intense mixing events. This would eventually lead to significant warming of the entire upper layer, mirroring conditions already observed in the Barents Sea. Therefore, future research should further examine the impact of this atmospheric-ice-ocean coupling on the pace of Arctic Atlantification.

Our experiments employ spatially and temporally averaged atmospheric forcing, which does not resolve the effects of synoptic-scale processes such as storm events. Research has shown that in the Eurasian Basin, strong storms are a key driver of sea ice decay, as they induce vertical ocean mixing that can cause a short-term surge in ice basal heat flux (Duarte et al., 2020; Graham et al., 2019). The averaged forcing smooths out these high-frequency events, preventing the model from capturing transient, storm-induced melting and mixing processes. However, the primary objective of this study was to quantify the independent and coupled effects of SIMW and ice-albedo feedbacks under a climatological mean state of the Arctic Ocean. This establishes a foundation for future investigations into the interactions among multiple feedbacks within the Arctic ice-ocean system. Future studies could explore the competition effects among storms, sea ice cover, and the surface freshwater budget. On one hand, storm-induced mixing disrupts stratification, transporting heat from deeper layers to the ice base and accelerating melt (Duarte et al., 2020). On the other hand, persistent summer SIMW release continuously rebuilds and maintains stratification, thereby suppressing upward heat transport (as demonstrated in this study).

Investigating this complex interaction is crucial for more accurately predicting the evolution of the Arctic sea ice-ocean coupling system against the backdrop of intensifying Atlantification.

Another limitation is that the model does not resolve two sub-grid scale features, under-ice false bottoms and deformed ice, which may introduce uncertainty into the quantitative assessments. First, the false bottoms can locally reduce ice melt by 7–8% over brief periods (Salganik et al., 2023a; Smith et al., 2023). This may lead to an underestimation of the stability provided by SIMW-induced stratification in our model, because the SIMW feedback strength quantified in this study represents only the effect of the SIMW-induced shallow mixed layer. If more stable features like false bottoms, observed to cover up to 21% of the area during the MOSAiC expedition (Salganik et al., 2023b), were included, the overall SIMW feedback could be stronger. Second, the model only considers level ice. However, the deformed ice, such as ridges and rubble that are not possible to represent in the 1D model, constitutes over 30% of the Arctic ice cover (Brenner et al., 2021). The deformed ice can melt significantly faster than level ice, and the SIMW-induced negative feedback that reduces bottom melt is less effective for them. This is primarily due to the complex topography of ridge keels interacting with ocean currents, which induces intense turbulent mixing (Skyllingstad et al., 2003). Salganik et al., (2023b) observed that summer average ocean-to-ice heat flux is approximately 17 W/m$^2$ under level ice but 65 W/m$^2$ under ridges. This implies that the integrated negative SIMW feedback across the Arctic is likely weaker than our model estimate, contrasting with the potential strengthening effect of false bottoms discussed earlier. Therefore, improving the model representation of the sub-ice features such as false bottoms and deformed ice processes is essential for accurately estimating the various feedback effects within the coupled Arctic ice–ocean system.

In the Arctic Ocean, melt pond formation is a process associated with SIMW. The melt ponds develop from the accumulation of snowmelt and surface sea ice melt, with their spatiotemporal morphology controlled by surface topography (Petrich et al., 2012; Polashenski et al., 2012; Webster et al., 2022). Melt ponds can temporarily retain SIMW, delaying its drainage into the ocean, a behavior somewhat analogous to the idealized "noMW" experimental scenario, which entirely prevents SIMW drainage to quantify its feedback but represents an extreme assumption. Observational evidence indicates that only about 10–15% of SIMW is retained on the ice surface in reality (Perovich et al., 2021; Smith et

al., 2025). Zhang et al. (2023) demonstrated that removing 20% of SIMW input results in only a 1% increase in summer sea ice melt, implying that the direct influence of SIMW retention caused by the ponds on ice cover is limited. In contrast, the albedo effect associated with melt ponds likely plays a more substantial role in sea ice melt. Melt ponds act as "windows" for solar radiation, significantly reducing surface albedo and enhancing radiation transmission into the upper ocean (Nicolaus et al., 2012), thereby accelerating ice melt. Consequently, future research can investigate more the interplay between SIMW retention and albedo effects to fully understand their combined impact on Arctic sea ice loss.

A recent study revealed the prevalence of low-salinity lenses in the marginal ice zones and shallow shelves of the Arctic Ocean, formed by localized intense sea-ice melt and river runoff. These lenses enhance summer sea-ice melting by trapping and concentrating solar radiation near the sea surface (Van Straaten et al., 2025). This effect contrasts sharply with the melt-inhibiting role of the SIMW in the deep basin identified in our study. The divergence underscores that the influence of surface freshwater layers on sea-ice cover differs fundamentally between shallow shelves and the deep basin. As sea ice retreat accelerates and the marginal ice zone shifts further into the central Arctic Ocean, the melt-enhancing effect of such lenses is expected to grow in importance. Therefore, the pan-Arctic integrated effect and regional variability of meltwater, or surface freshwater layers more broadly, warrant further investigation. In addition, this study primarily focuses on quantifying SIMW and ice-albedo feedbacks during summer, as well as the effects of SIMW on the ice-ocean system in the subsequent winter. However, the winter sea-ice formation process also involves several feedbacks that require more in-depth study. These include the ice production-entrainment feedback, where brine rejection during sea ice formation brings heat from deeper layers to the surface, melting some of the initially formed ice and inhibiting further production (Goosse et al., 2018; Martinson and Iannuzzi, 1998), and the ice growth-thickness feedback, where thin sea ice grows more rapidly than thick ice due to its higher thermal conductivity (Bitz and Roe, 2004), a phenomenon recently observed in the Arctic Ocean (Lin et al., 2022). Indeed, evidence of these two winter feedbacks is present in our experiments. For example, the CTRL run show less sea ice production during winter at the weakly stratified stations (NS2-NS5) compared to other stations (Figure 5b), which is primarily driven by the ice production-entrainment feedback. Similarly, greater sea ice growth in the CTRL run than in the noIA run (Figure 5b) aligns with the ice growth-thickness feedback.

The primary contribution of this study lies in clearly demonstrating the independent and interactive effects of SIMW and ice-albedo feedbacks on the ice-ocean system under mean-state conditions, thereby providing a useful conceptual framework for future quantification of other coupled feedback mechanisms. We suggest that future research should build upon this foundation by integrating three-dimensional models to incorporate advective processes; employing high-resolution forcing to resolve storm-induced variability and transient events; refining the representation of sub-ice topography and melt pond effects to improve feedback estimates. Additionally, extending investigations to inter-annual timescales and incorporating winter-specific feedbacks (e.g., ice production-entrainment and growth-thickness mechanisms) will be crucial for projecting the changes of Arctic Atlantification and sea ice loss under evolving climate forcings.

## 6 Conclusions

This study employs a one-dimensional coupled sea ice–ocean model to quantify two key feedbacks under a climatological mean state. Through a series of decoupling experiments, we assessed the independent roles of SIMW and ice-albedo feedbacks in summer melting and their subsequent effects on winter processes. Although our simplified one-dimensional framework does not resolve advective processes or sub-grid scale features, it successfully reproduces the observed evolution of key physical quantities in the central Arctic on seasonal timescales. To the best of our knowledge, this is the first study to quantify the above feedback effects, both individually and in combination, in the Arctic ice-ocean system. Therefore, this work provides a critical conceptual foundation and a quantitative benchmark for understanding these coupled feedbacks. The main conclusions are as follows:

1. During the melting season, the negative feedback induced by SIMW is substantial, reaching nearly half the strength of the positive ice-albedo feedback. Specifically, the SIMW feedback reduces total summer ice melting by 19% ($\gamma_{MW} = -0.19$), while the ice-albedo feedback amplifies it by 41% ($\gamma_{IA} = +0.41$);

2. These two feedbacks are nonlinearly coupled. The strength of the SIMW feedback is highly dependent on the ice-albedo feedback, as its efficacy drops by more than half (to $\gamma_{noMW} = -0.09$) when ice-albedo

processes are suppressed. In contrast, the ice-albedo feedback is only marginally enhanced (to $\gamma_{noIA}$ = +0.46) when SIMW effects are removed, underscoring its dominant role in the system;

3. There is a seasonal compensation mechanism for sea ice: enhanced summer ice melt can, paradoxically, lead to increased ice formation during the subsequent winter in strongly stratified regions. This occurs because greater summer melt results in a thinner ice cover and expanded open water areas by the start of winter, which collectively enhance the ocean heat loss to the atmosphere and accelerate the freeze-up in the early winter;

4. In the weakly stratified Nansen Basin, SIMW plays a critical role in maintaining stratification, which insulates the sea ice from the heat of the AW below. The absence of SIMW in this region triggers intense winter vertical mixing and ocean-to-ice heat flux.

## Data availability

The Ice-Tethered Profiler data were collected and made available by the Ice-Tethered Profiler Program based at the Woods Hole Oceanographic Institution (Krishfield et al., 2008; Toole et al., 2011) and are available at https://doi.org/10.7289/v5mw2f7x (Toole et al., 2016). NCEP/DOE Reanalysis II data provided by the NOAA PSL, Boulder, Colorado, USA (https://psl.noaa.gov). The World Ocean Database 2018 and World Ocean Atlas 2023 are available through the National Centers for Environmental Information (NCEI) archives (https://www.ncei.noaa.gov/products/). The monthly ice thickness and concentration are permanently deposited to NSIDC, https://nsidc.org/data/.The 1D model configuration, parameters and forcing fields used in this study are stored at https://github.com/HaohZhang/1D-model.

## Author contributions

HH conducted the experiments and model validation, analyzed the data, and drafted the initial version of the manuscript. AS and CY designed the experiments. HH and XB Conducted the 1D model construction and code modification based on the MITgcm. HH, AS, XB and CY all participated in the review and editing of this paper.

## Competing interests

Neither of the authors has any competing interests.

## Acknowledgments

This work was funded by the National Natural Science Foundation of China (Grant No. 42276254). Haohao Zhang was also supported by the scholarship from China Scholarship Council (Grant No. 202306710085).

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
