# Peer review of "Quantifying the interplay of Sea Ice Meltwater and Ice—Albedo"

_EGUsphere, 2025_

## Referee Comment (RC2)

General Comments:

The overall objective of the study is to *quantify the independent effects of meltwater and ice-albedo feedbacks on the Arctic ice-ocean system*. Specifically, the work proposes to anser three questions:

- *To what extent do the meltwater and ice-albedo feedbacks influence ice melting during summer?*
- *What is the contribution of the meltwater (ice-albedo) feedback if the ice-albedo (meltwater) feedback is not involved?*
- *What are the regional differences in the impact of meltwater on the ice-ocean system?*

These research objectives are within the scope of The Cryosphere and, to the best of my knowledge, are novel. The difficulty of the research arises from the strong coupling between the feedbacks of meltwater and ice albedo, which prevents a clear differentiation of the contribution of each. To solve this problem, the authors propose a methodology based on a series of simulations where feedbacks are selectively activated. Independent contributions are then encoded in feedback factors. I consider the methodology appropriate, implemented in a rigorous way and well documented. The supplementary material reinforces its validity with an extensive comparison between the model and observations. However, I believe that its description would benefit from the inclusion of the 1D dynamic equations either in the main text or in the supplementary material.

The presentation and discussion of the results raised my main concerns with the manuscript. Model results are quantitatively presented but, in main opinion, poorly interpreted from the physical point of view. In some sections (see below), the explanation of some model results would require a qualitative interpretation or a physical hypothesis. The quantities are only indicated and at most justified with references or general assessments. Brief physical explanations would be very informative.

I consider that a revised version of the manuscript could be suitable for publication in The Cryosphere.

Specific Comments:

L 55: *In winter, brine rejection caused by ice formation leads to the upward entrainment of heat from the NSTM and AWW and impedes winter ice formation consequently (Smith et al., 2018; Steele et al., 2011; Timmermans et al., 2017), which is a negative feedback known as the ice production-entrainment feedback (Goosse et al., 2018)*. My understanding is that the ice production-entrainment feedback inhibits vertical heat transfer, promoting more ice production. could you confirm this point?

L 72 *Zhang et al., (2023) demonstrated that the removal of meltwater increases ice melt by 17%* Could you add briefly the physical mechanism proposed by Zhang et al. to justify this effect?

L 75 Indicate the differentiated aspects considered in the work between meltwater and icea albedo feedbacks. There is a strong relationship between both and it is not clear their intrinsic and differentiated components.

L 102 I would suggest to include the mathematical expression of the 1D model.

L 116: A boundary condition is always required to get a particular solution to the differential equation. Please, clarify the sentence *No boundary condition is applied at the bottom of the model.*

L 175 *Based* ON *the feedback factor (γ) framework proposed by Goosse…*

L 177 *Total response* by Total Response

L 179  (Reference Response, RR)

Figure 3. Please define the acronyms in the figure caption. The figure is cited before they are defined in the text.

L 264-265 *the brine rejection process during winter creates higher ocean-ice heat flux, which in turn inhibits ice formation*. Could you briefly clarify the mechanism of this? As I understand it, brine rejection contributes to the formation of the cold halocline that isolates the upper Arctic layers from the warm waters of the Atlantic. It would favor ice formation in this case. Could you clarify the assessment?

L 270 *the removal of meltwater results in a 0.19 m increase in ice formation compared to the CTRL run*. Why? How do you physically interpret this result?

L 273-275 *The noMWIA run also shows less ice formation compared to the noIA run (Figure 5b) in station NS2, NS4 and NS5, but no sea ice melting is observed during winter, further demonstrating the importance of meltwater in weakly stratified regions*. What is the physical explanation for this result?

L 277 *This suggests that sea ice retreat during summer can promote ice formation in winter*. Can you give a physical reason based on your model? Is it just a geometric effect of having more ocean surface available to freeze?

L 291  *the NSTM, because the heat absorbed by the ocean can mix sufficiently within the upper 30 m.* This explanation would disagree with observations. The  NSTM layer can be observed at this depth in the Canada Basin (Jackson 2010). Provide an explanation to the discrepancy between the model and observations.

L 297 …agree well with the observed values…

Figure 6.Add month labels to the figures corresponding to station BS5, similarly to those of NS2

Figures 6 a,e,I,m,c,g,k and o. The discontinuity in the MLD seems an artifact of the surface tracking algorith due to the appearance of a new water mass, the meltwater. A gradient tracking algorithm could identify this as the bottom of the mixed layer. There is no discontinuity in the rest of simulations because the meltwater input is cancelled. Could you clarify this point?

L 305 How do you explain this big impact of meltwater in the mixed layer depth during the freezing season? Is there any observational or 3D model result that corroborates it?

L 390 *This is a process similar to the noMW run, where less meltwater entering the ocean accelerates sea ice melting.* Meltwater from ponds, sooner or later, flows into the Ocean. This might slow or delay the formation of a meltwater layer, but probably not inhibit it like in the study. Could you comment?

L 455 *it delays winter ice melting by slowing surface cooling*. Should be warming instead cooling?

---

## Author Response (AR1)

**Response to Reviewer #1**

**I find that this is an interesting study that provides insights into the role of seasonal sea ice melt (water) on the evolution of sea ice and the upper ocean across the deep Arctic basin. It uses a 1D coupled ice-ocean model that has been used in earlier somewhat similar studies. It nicely examines the sensitivity of the coupled system to seasonal sea ice meltwater release and ice-albedo feedback. It is also generally well written. And my only "objections" in this first round are just that authors should make clear of the "caveats" of such a simplified approach when extrapolating the results. I elaborate on these points below. Beyond that I think that this work fits perfectly in the scope of TC, and is publishable with minor revisions given the authors also reflect on the points I raised below in the Discussion and Conclusions of the revised manuscript and place the work properly in context of references given.**

**Answer:** Thank you very much for your thorough review and constructive feedback, which significantly enhanced the quality and clarity of our manuscript, especially by placing our research in a broader real-world context. Based on your comments and suggestions, we made the following key revisions: Added discussions on the model's limitations and comparisons with other observational studies; reran all experiments with snowfall and zero external freshwater forcing; Provided detailed descriptions of the ocean model, sea ice model, and albedo parameterization in the Supplementary Information; Optimized figures in both the main text and Supplementary Information; Further refined the Abstract and Conclusion sections. We now answer each of your comments point by point below. For clarity, **your comments are in bold black**, our responses are in blue, and *the revised text in the manuscript is presented in italic*. All authors approved these changes. In this response, figures are labeled as Fig. R1, Fig. R2, etc.

**Generic comments**

**Q1.** **The Introduction could benefit from noting on the very small-scale meltwater layers that form below sea ice (Smith et al., 2023; Salganik et al., 2023a), and aren't necessarily resolved by the model setup, and are perhaps more stable than in the model setting(?). Salganik et al. (2023a) also provides a direct estimate of the effect of such meltwater layers on the summer melt of (level) sea ice, albeit from very short-term observations, which might be interesting to compare to. Further Perovich et al (2021) and Smith et al. (2025) discuss the fate of sea ice meltwater based on direct observations, and latter indicates that 10% of the meltwater does not enter the ocean, but that would have a small impact on your results, would it? I would think the findings from these recent studies would be good to introduce and used to place this work better in context.**

**Answer:** Thank you for these valuable suggestions, which greatly enriches the research background of this paper. We revised the Introduction and Discussion sections to incorporate these recent studies. The revised Introduction now includes the effects of

the very small-scale meltwater layers. The revised Discussion now compares these results with our model findings and discusses the limitations of our model setup.

[revised manuscript text omitted]

**Q2.** **From what I understand, the work conducted assumes all ice is level ice, and this is obviously not the case in reality. Deformed ice (ridges, rubble etc.), can easily have an areal fraction of >30% (e.g. Brenner et al., 2021, see their Suppl. Material), and this is known i) affect (limit) the spreading of sea ice meltwater and that ii) ridges melt much faster than level ice (e.g. Salganik et al., 2023ab) - so I would perhaps at least note this fact and be somewhat careful when extrapolating the results, as the "meltwater" layer effect likely applies only to a fraction of the ice cover - not the full ice cover - given thicker ice might melt much more rapidly, so the overall effect might be much less than proposed here.**

**Answer:** Thank you very much for raising this key question. Yes, our model only considered level ice. We revised the Discussion and pointed out that the sea ice meltwater feedback may be invalid for deformed ice in the Arctic Ocean and explained the reasons based on literature (see the "Revision" section as shown in Q1).

**Q3.** **Lines ~40-50: I am not an oceanographer, but I find this "freshwater" budget presented here, somewhat misleading. When you look at the salinity profiles you use as initial conditions, there is a large freshwater inventory already before sea ice melt onset, which from my understanding is primarily meteoric water (river + net P/E). This meteoric inventory is order of 10+ meters (e.g. Bauch et al., 1995; Dodd et al., 2012) or even more in places, that has accumulated over time and to me its this freshwater component that sets the scene for the overall layering in the Arctic Ocean, not the seasonal sea ice melt. It's the former (aided by sea ice formation) that creates the halocline that is the barrier between AW and the surface. To my understanding sea ice melt creates the seasonal shallower mixed layer. And with the exception on parts of Nansen basin that do not have effective sources of river runoff, as you also note. Sea ice formation also plays a crucial role in distributing this freshwater. Seasonal sea ice meltwater is then probably more important for the NSTM, given how deep solar radiation can penetrate the ocean. Anyway, I felt that this part needs some clarification in this regard.**

**Answer:** Thank you for your suggestion. We fully agree with your perspective. In the Arctic Ocean, meteoric water serves as one of the primary contributors to the long-term freshwater balance and stratification, alongside sustained freshwater inputs from river runoff and inflows through straits. These freshwater sources establish the precondition for upper ocean stratification, which exhibits clear regional variability—specifically,

freshwater content gradually decreases from the Canada Basin to the Eurasian Basin, accompanied by a corresponding weakening of stratification. These differences in stratification, in turn, lead to spatially varying effects on sea ice melt/freeze processes. In contrast, the sea ice melt/freeze cycle plays a more dominant role in driving seasonal mixed-layer variability across the Arctic Ocean, as it involves substantial short-term injection and extraction of freshwater in the surface layer. We revised the Introduction to more clearly state that the meteoric water is a major factor in the long-term freshwater budget and stratification of the Arctic Ocean, while the sea ice melt–freeze cycle plays a more important role in seasonal changes.

**Revision:** Discussion, lines 34-48: *On the interannual scale, the freshwater input from meteoric water (e.g., net precipitation and river runoff) governs the freshwater balance and stratification of the Arctic Ocean (Serreze et al., 2006), as it significantly contributes to the freshwater content-equivalent to a freshwater layer approximately 10 meters in the upper Arctic Ocean (Bauch et al., 1995; Dodd et al., 2012). On the seasonal scale, the fresh surface mixed layer is largely influenced by the sea ice melting/freezing cycle (Hordoir et al., 2022; Morison and Smith, 1981; Peralta-Ferriz and Woodgate, 2015; Polyakov et al., 2013). Each summer, sea-ice melt contributes approximately 1.2 meters of freshwater to the upper Arctic Ocean (Haine et al., 2015), which is markedly less than the long-term meteoric freshwater inventory of ~10 meters. However, this input is released intensively over months (~11,300 km³ during summer) (Haine et al., 2015), which can result in relatively thin sea-ice meltwater (SIMW) layers in the upper ocean and rapidly establish a shallow summer mixed layer (Hordoir et al., 2022; Peralta-Ferriz and Woodgate, 2015; Smith et al., 2023). Although river runoff also contributes a substantial freshwater during summer to the Arctic Ocean (~4,200 km³), it tends to remain confined to the coastal regions on seasonal scales (Osadchiev et al., 2020, 2021), with its subsequent transport pathways into the deep basin influenced by atmospheric circulation regimes (Wang et al., 2021a).*

**Q4. Albedo - I think the main text should include a brief description of how the model treats sea ice albedo (also add a panel of albedo in Figure 4), with reference to the Appendix. Given that you try to single out the effect the of albedo, this should be more thoroughly explained in Section 2, and showing it in a panel in Figure 4 would also be very helpful to assess the albedo evolution in the model.**

**Answer:** Thank you for your suggestion. In Section 2.1 of the manuscript, we added a few sentences to describe the ice/snow albedo parameterization in the model, and we also provided the detailed of albedo parameterization in the Supplementary Information (In the revised Supplementary Information, we included more equations of the one-dimensional model). We also added the albedo time series into Figure 4 in the revised manuscript (see Figure R1 below).

**Revision:** Section 2.1, lines 158-165: *The model parameterizes the sea ice/snow albedo as a composite value that integrates contributions from both bare ice and overlying snow. Ice albedo is formulated using an exponential decay function dependent on ice*

*thickness. Snow albedo incorporates both thermal state and aging effects, where fresh snow albedo varies linearly with surface temperature between cold and warm limits, while aged snow albedo decays exponentially with snow age toward an asymptotic old-snow value. The combined surface albedo is then calculated by weighting ice and snow albedos through an exponential attenuation function based on snow depth (detailed description of the sea ice model can be found in Section 1 of the SI).*

[Figure]

**Figure R1.** Modeled (a)-(b) sea ice thickness (SIT); (c)-(d) sea ice concentration (SIC); (e)-(f) ice/snow albedo; (g)-(f) net ocean shortwave heat flux (Fsw), negative values represent heat entering the ocean. (i)-(j) sea ice meltwater flux ($F_{MW}$), negative values represent freshwater entering the ocean. The left column is the result of station BS-5 and the right column is the result of station NS-2. The dashed lines perpendicular to the X-axis represent the first freezing day of each experiment (defined as the first day on which the ice thickness growth rate surpasses 0.1 cm/day).

**Q5.** **I would like that you assess better how the above factors affect the generalization of your results in the revised manuscript. The Conclusion now lacks any appreciation of how the "real world" might differ from the simplified model experiments, nor compare to earlier findings, I would find that to be appropriate for the benefit of the reader (its also rather superficial in the Discussion as well). It would also be useful to advise future studies what could potentially be done better/different.**

**Appreciate the illustrations esp. Figs 2 and 10. Nice work.**

**Answer:** Thank you for your suggestions and for your appreciation of our work. In addition to the discussions, we added based on your comments in Q1-Q4, we included

more discussions and suggestions for future work based on your other specific comments. For details, please see our responses to the specific comments below.

**Specific comments**

**Q6,** **L101 - in this section (someplace in the main text) I think it would be good to briefly note on how the surface albedo of ice/ocean is treated in the model, especially for the ice. This is probably important when considering the relative impact of meltwater or albedo.**

**Answer:** Thank you for your comment. Please refer to Q4 of the Generic comments.

**Q7,** **L143 - snow depth, how does this evolve over time, does it reappear in fall? I did not quite understand that, but perhaps you can add a sentence to clarify. While not so important for melt, snow will definitely also impact ice growth.**

**Answer:** We sincerely thank you for raising the critical point. We fully agree that while snow melts completely in summer and thus does not affect our core conclusions on summer feedbacks, it re-accumulates in autumn and winter, playing an essential role by insulating the sea ice from the cold atmosphere and thereby regulating winter ice growth (Sledd et al., 2024). To enhance the physical realism of our model experiments, we incorporated the snowfall and rerun all numerical experiments. The results indicate that incorporating winter snowfall reduces winter ice formation by approximately 18 cm compared to previous experiments (see Figure R2 below). The results presented in the main text now are new results, incorporating winter snowfall. We clarified in the revised manuscript that our model includes winter snowfall.

[Figure]

**Figure. R2.** Modeled time series of sea ice thickness with and without winter snowfall.

**Revision:** Section 2.3, lines 221-226: *The model also accounts for winter snowfall, as snow cover can insulate the sea ice from the cold atmosphere and thereby regulate winter ice growth (Sledd et al., 2024). Specifically, snowfall in the model begins on October 1st and results in a snow layer approximately 0.19 meters on the sea ice surface by the end of the model period (next April 30th). This value is consistent with both the initial snow condition and the satellite-observed basin-averaged Arctic snow thickness for April (Kacimi and Kwok, 2022; Kwok et al., 2020).*

**Q8,** L145-160, Fig 1 & 2. Grouping of stations and initial ocean conditions - what caught my eye is the profile for NA-5, this seem very different than anything around in (NA or AM), and thus wonder, have you checked other profiles in this area to see whether this is a recurring type of profile. I would have based on location usually expected something more like the rest of the NA or AM profiles. NS-1 is what I would say typical Transpolar drift profile you find in AM/NA, and here using the choice of geography to categorise the station is a bit misleading. But I assume a single station does not make any difference to the final interpretation for the regions.

**Answer:** Thank you for this comment. We checked additional in-situ profiles from the vicinity of station NA5 (collected in April and May 2011), which show similar characteristics to those of NA5 (see Figure R3 below). In fact, given the geographical proximity of NA5 to the Amundsen Basin, it is reasonable that its water mass properties tend to resemble those of the Amundsen Basin. We revised section 2.2 to add a brief explanation regarding the atypical characteristics of stations NA5 and NS1 within their respective groups.

In addition, we note that in the original manuscript, Figure 2 used different x-axis ranges for different basins, which may cause confusion for readers. In the revised version, we revised Figure 2 to use consistent x-axis ranges across all basins to facilitate clearer comparison (see Figure. R4 below).

[Figure]

**Figure. R3.** Locations and profiles around station NA5. These profiles taken from the World Ocean Database 2023 (WOD23, Mishonov et al., 2024)

[Figure]

**Figure R4.** Observed salinity (left column), temperature (middle column) and buoyancy frequency (right column) from observational profiles between 2011 and 2023 in the Arctic Ocean, used as initial profiles in the model simulations. To capture representative characteristics of each region in the Arctic Ocean, these profiles were grouped geographically into four categories: (a)-(c): Beaufort Sea (BS); (d)-(f): North of the Amerasian Basin (NA); (g)-(i): Amundsen Basin (AM); (j)-(l): Nansen Basin (NS). The time of each profile is shown in the color scale at the right-hand side. In order to emphasize the regional differences of the upper stratification, we show only the upper 300m. In fact, in the model we used the upper 700 m of data as the initial field.

**Revision:** Section 2.2, lines 176-186: *To capture representative characteristics of each region in the Arctic Ocean, these profiles were grouped geographically into four categories: the Beaufort Sea (BS), the Northern Amerasian Basin (NA), the Amundsen Basin (AM) and the Nansen Basin (NS). The buoyancy frequency profiles show that ocean stratification gradually weakens from the Pacific-influenced sector towards the Atlantic-influenced sector (Figure 2). Within each group, the profiles generally exhibit consistent features, only two individual stations display slight deviations from their group's typical patterns. Specifically, station NA-5, which is located in the NA but close to the central Arctic, exhibits higher salinity and a warmer AWW layer, with its profile more similar to those from the AM. Meanwhile, station NS-1 exhibits characteristics closer to those of the AM region, with notably stronger upper-ocean stratification than stations NS2–NS5. Overall, these 20 profiles demonstrate the diverse stratification characteristics present throughout the Arctic Basin.*

**Q9,** L167-169 - Somewhat confusing, first you say that you use "two subregions" but the map shows one single area with the red line in the map. Is the forcing averaged over the whole area and that is used as forcing at each site.

**Answer:** Thank you for highlighting this confusing description. Yes, the forcing averaged over the whole area and that is used as forcing at each site

**Revision:** Section 2.3, lines 216-221: *The atmospheric forcing for the 1D model is derived from daily averages of atmospheric variables (10 m wind speed, 2 m air temperature, specific humidity, sea level pressure, downward longwave and shortwave radiation) based on the National Centres for Environmental Prediction-Department of Energy (NCEP-DOE) Reanalysis 2 (Kanamitsu et al., 2002). We compute a climatological daily average (2011–2023) by spatially averaging these variables over the entire region enclosed by the red boundary in Figure 1, which uniformly averaged forcing is applied to each experiment.*

**Q10,** L170-171 - Later you state this external FW flux is very small compared to sea ice meltwater. If you run the experiments with zero external freshwater are the results any different? I also find that this term is a bit awkward, given that any advective FW (relative to ice motion) would not be only at the surface, but in the whole upper water column rather than a continuous flux at the very surface. The magnitude when averaged over the Arctic maybe is as such reasonable, but the way its implemented not.

**Answer:** Thank you for this comment, we agree it is necessary to compare the experimental results with and without external FW flux. We initially included the external FW flux aims to ensure a complete freshwater budget in the 1D model, as omitting it could potentially lead to an overestimation of the role of sea ice meltwater. Following your suggestion, we conducted a sensitivity experiment without external freshwater forcing for all stations. Results show that experiments with zero external freshwater slightly enhance the strength of the meltwater feedback (from -0.19 to -0.2) and deepen the mixed layer depth, but overall have little impact on the outcomes (Figure. R5 and Figure. R6). We compared the experiments with external freshwater forcing values of 0 km³/yr, 1200 km³/yr (the recommended value in this paper), and 6400 km³/yr (the excessive value) together, and included these results in the Supplementary Information.

    Regarding the implementation of the external FW flux as ocean surface flux, we acknowledge the limitation of representing external FW flux as a surface flux. This approach is widely adopted in ocean models (e.g., MITgcm, ROMS) for surface boundary conditions to handle processes such as precipitation, evaporation, river runoff, and sea ice melt/freeze, typically implemented as a virtual salt flux.

[Figure]

**Figure. R5.** Box plots illustrate the four feedback factors across different stations, with external freshwater forcing values of (a) 0 km³/y, (b) 1200 km³/y and (c) 6400 km³/y.

[Figure]

**Figure. R6.** Time series of the mean MLD for each basin, obtained from simulations using external freshwater forcing values of 0 km³/y, 1200 km³/y and 6400 km³/y.

**Q11,** L171 - how does this value compare typical sea ice meltwater fluxes? And if I understand correctly this is never changed in the four experiments?

**Answer:** Thank you for your comment. As shown in Figure R1 under **Q4**, the maximum sea ice meltwater flux in the control run during summer is approximately $2\times10^{-7}$ m/s, which is far greater than the external freshwater input value ($3.92 \times 10^{-9}$ m/s). Yes, this value is applied to all experiments.

**Revision:** Section 2.3, Line 234: '…*which is applied to all experiments…*'

**Q12-17,** L181 - do you mean "sea ice meltwater discharge" or does this also mean the "external freshwater" is zero, please clarify;

    L205 - please specify what meltwater;

    L184 - you mean "sea ice meltwater flux" - just to be consistent in the use of terms;

    L205 - please specify what meltwater;

    L191 - specify whether its only sea ice meltwater is set to zero, or also external meltwater. Just to be sure. Thank you.

    L444 - "sea-ice meltwater" - again specify for the benefit of the reader (applies to the whole manuscript).

**Answer:** Thank you for your comments and we would like to put these similar comments together for a single response here. In this context, 'meltwater' refers specifically to 'sea ice meltwater'. For clarity, we have replaced all 'meltwater' with 'sea-ice meltwater (SIMW)' throughout the manuscript, including in the title.

**Q18,** L187-188 "results are presented in Section 3.

**Answer:** Thank you for your careful review. We revised this sentence.

**Q19,** L226-227 - I assume SWpenetrate is the energy transmitted through the ice to the ocean? I assume the more correct term is then "transmitted".

**Answer:** Thank you for your suggestion. We revised Figure 3 and its caption (see Figure. R7 below).

[Figure]

**Figure R7.** Schematic representation of the four experiments in this study. …… SWtransmitted (shortwave radiation transmitted through sea ice to the ocean),……

**Q20,** L230 - Fig. 4. - I wonder if it would be useful to show the panels in the following order, SIT, SIC, Fsw, SFW, given the former two drive the latter. and could add a weak horizontal line at zero in current panel d. Also in panel d, indicate the "external freshwater flux". And the legend would preferably be placed in the top panel.

**Answer:** Thank you for your suggestion. Please refer to Q4 of the Generic comments.

**Q21-22, L236 - change to "initial ocean profiles"**
      **L238 - change to "impact of initial ocean profiles"**

**Answer:** Thank you for your careful review. We revised these two sentences.

**Q23, L238-240 - In light of this, I would suggest you add station NS-2 to figure 4, so the left-hand side is for BS-5 and right hand side for NS-2.**

**Answer:** Thank you for your suggestion. We revised the Figure 4 (Please refer to Q4 of the Generic comments).

**Q24, L239-240, the word "condition"*2 can be deleted.**

**Answer:** Thank you for your careful review. We revised this sentence.

**Q25, L280 - Fig 5. I would think it would be much simpler if you grouped the stations by the same color as in Fig 1. I think the within group variation is only large for the NS stations (and perhaps the one AM station), so it would probably be enough to use fewer colors. Applies also to Fig 7 and 9, especially in 9 they dots are overlain and does not make a difference showing each station with a different color in my opinion.**

**Answer:** Thank you very much for this suggestion. We revised Figure 5, 7 and 9 (see Figure R8-R10 below).

[Figure]

**Figure R8.** Box plots illustrating the (a) ice thickness changes during the melting season and (b) freezing season across different Stations in different types of experiments. Each box plot shows the median, interquartile range, and potential outliers (points marked with red plus sign). All points are the results of experiments with initial SIT of 2 m.

[Figure]

**Figure R9.** Box plots illustrating the mean (a) MLD in summer and (b) winter across different Stations in different types of experiments. Each box plot shows the median, interquartile range, and potential outliers (points marked with red plus sign). All points are the results of experiments with initial SIT of 2 m.

[Figure]

**Figure R10.** Box plots illustrating the four feedback factors across different stations. If the $\gamma$ is a positive (negative) value, it represents positive (negative) feedback. All points are the results of experiments with initial SIT of 2 m. The blue star and red star represent stations BS5 and NS2, respectively. Ice-albedo feedback Factor: $\gamma_{IA}$; SIMWfeedback Factor: $\gamma_{MW}$; Net SIMW feedback factor: $\gamma_{noIA}$; Net ice-albedo feedback factor: $\gamma_{noMW}$.

**Q26,** L320 - Fig 6 - a closer look at the typical types of ocean profiles (Fig. 2), it would be informative to show here also one station that is "in between", and to me this would perhaps be AM-1 or AM-2. In terms of less surface freshening than BS-2, but also a different initial heat content than BS-2. I would think this is useful to have in the main text.

**Answer:** Thank you for your comment. We fully agree that incorporating an intermediate state station would provide readers with a clearer illustration of regional differences. We revised this Figure (see Figure R11 below) and added a few sentences to describe the station AM2.

**Revision:** Section 3, lines 383-386: *At station AM2, which has intermediate stratification weaker than BS5 but stronger than NS2, removing SIMW increased the maximum MLD from 49 meters in the CTRL run to 92 meters in the noMW run, deepening by approximately 40 meters. Although the vertical mixing is more pronounced compared to station BS5, it still cannot reach the core depth of the AWW layer (Figure 5i).*

*And lines 388-390: At station AM2, the MLD in the noMWIA run only deepened by about 10 meters compared to the noIA run (Figure 5k and l), much smaller than the 40-meter difference between the noMW run and the CTRL run.*

[Figure]

**Figure R11.** Simulated vertical profiles of ocean temperature and salinity over time at stations (a)-(h) BS5, (i)-(p) AM2, and (q)-(x) NS2. For each station, the top row displays temperature, and the bottom row displays salinity. From left to right, the columns correspond to the CTRL run, noMW run, noIA run, and noIAMW run. Cyan dots in each panel represent the mixed layer depth. Note that the vertical depth scales are not consistent across stations.

**Q27-28,** L386 - Stating "is reliable" sounds over convincing yourself. Rather, I would rather phrase this in some more insightful way, how you can "tease out" the possible contribution of different factors. But I would still have some doubts that using e.g. a uniform forcing all across the Arctic, might not be representative in regions with much atmospheric activity and strong synoptic events, which are taken out by averaging the forcing? Then especially again the Atlantic sector., e.g. Graham et al. (2019).
L400-401 - Duarte et al. (2020) point towards synoptic events being important in the Nansen basin region, how does your "arctic wide averaged forcing" mean for this type of single-events? And that single-event ocean heat fluxes (up to 400 Wm2, see Duarte et al. 2020 and references therein) are important in this region (see also Graham et al., 2019).

**Answer:** Thank you for your suggestions. We would like to answer these two comments together because they both relate to the mean forcing field in the model and real-world synoptic events. In the revised manuscript, we completely rewritten the Discussion section. We removed highly subjective phrases like "is reliable" and 'well-validated'. As answers of previous comments, we discussed potential impacts on our model results, such as from deformed ice and subglacial false lows.

We added a paragraph in the Discussion section to explain that the main purpose of this paper is to study the relative strength of sea-ice meltwater feedback and ice-albedo feedback under average conditions. The paragraph also highlights the model's limitation in representing weather-scale events and provides some suggestions for future research.

**Revision:** Discussion, lines 606-620: *Our experiments employ spatially and temporally averaged atmospheric forcing, which does not resolve the effects of synoptic-scale processes such as storm events. Research has shown that in the Eurasian Basin, strong storms are a key driver of sea ice decay, as they induce vertical ocean mixing that can cause a short-term surge in ice basal heat flux (Duarte et al., 2020; Graham et al., 2019). The averaged forcing smooths out these high-frequency events, preventing the model from capturing transient, storm-induced melting and mixing processes. However, the primary objective of this study was to quantify the independent and coupled effects of SIMW and ice-albedo feedbacks under a climatological mean state of the Arctic Ocean. This establishes a foundation for future investigations into the interactions among multiple feedbacks within the Arctic ice-ocean system. Future studies could explore the competition effects among storms, sea ice cover, and the surface freshwater budget. On one hand, storm-induced mixing disrupts stratification, transporting heat from deeper layers to the ice base and accelerating melt (Duarte et al., 2020). On the other hand, persistent summer SIMW release continuously rebuilds and maintains stratification, thereby suppressing upward heat transport (as demonstrated in this study). Investigating this complex interaction is crucial for more accurately predicting the evolution of the Arctic sea ice-ocean coupling system against the backdrop of intensifying Atlantification.*

**Q29, L387-393 - As noted in the generic comments the work of Perovich et al and Smith et al are relevant mention here. Also the fact that with melt ponds you significantly increase the transmission of solar radiation to the ocean and not only absorption to ponds/ice and decrease the albedo. Refer to Nicolaus et al. (2012).**

**Answer:** Thank you for your comment. In the discussion, we added some sentences that cited these studies and mentioned that the melt ponds can change the surface albedo.

**Revision:** Discussion, lines 640-653: *In the Arctic Ocean, melt pond formation is a process associated with SIMW. The melt ponds develop from the accumulation of snowmelt and surface sea ice melt, with their spatiotemporal morphology controlled by surface topography (Petrich et al., 2012; Polashenski et al., 2012; Webster et al., 2022). Melt ponds can temporarily retain SIMW, delaying its drainage into the ocean, a behavior somewhat analogous to the idealized "noMW" experimental scenario, which entirely prevents SIMW drainage to quantify its feedback but represents an extreme assumption. Observational evidence indicates that only about 10–15% of SIMW is retained on the ice surface in reality (Perovich et al., 2021; Smith et al., 2025). Zhang et al. (2023) demonstrated that removing 20% of SIMW input results in only a 1% increase in summer sea ice melt, implying that the direct influence of SIMW retention caused by the ponds on ice cover is limited. In contrast, the albedo effect associated with melt ponds likely plays a more substantial role in sea ice melt. Melt ponds act as "windows" for solar radiation, significantly reducing surface albedo and enhancing radiation transmission into the upper ocean (Nicolaus et al., 2012), thereby accelerating ice melt. Consequently, future research can investigate more the interplay between SIMW retention and albedo effects to fully understand their combined impact on Arctic sea ice loss.*

**Q30, L396-423 - In general this seems to point me to the fact that advective terms can be very important in the Nansen basin case of this work? Omitting those, could possibly distort the results presented here signficantly? Given that ice is always transported into this region with the Transpolar Drift often replacing melted ice, providing more potential for meltwater sources, and heat is also continuously also transported with Atlantic boundary current "replacing" lost ocean heat.**

**Answer:** Thank you for your comment. We agree that the role of advection in this region is very important, as the inflow of Atlantic water significantly influences the heat content in this region. Our model shows even under the condition without the heat supplementation, after removing meltwater, the station in the Nansen Basin exhibited intense vertical mixing and substantial ocean-to-ice heat flux during winter. This demonstrates that the heat already stored in the Atlantic warm water, once it breaks through the upper stratification, is sufficient to counteract the cold atmospheric conditions in winter and melt sea ice. With heat supplementation, these effects would be more pronounced.

Additionally, the loss or accumulation of sea ice or surface freshwater due to advection is largely influenced by atmospheric forcing (which will be clarified in the revised manuscript shown below). Our advection-free experiment primarily represents an average state. Furthermore, our experiments involving the removal of meltwater and maintaining constant sea ice provide valuable insights into the potential impacts of atmospheric forcing on sea ice or freshwater loss/accumulation, as these experiments inherently alter the surface freshwater sources. We added two paragraphs in our revised manuscript to elaborate on the above points in detail.

**Revision:** Discussion, lines 565-586: *Nevertheless, the influence of advection in the Arctic ice-ocean system cannot be overlooked. This is particularly evident in the western Eurasian Basin, where AWW inflow and its associated heat transport significantly impacts the local sea ice-ocean system (Polyakov et al., 2017). The West Spitsbergen Current delivers a heat flux of up to 200 W m⁻² into the Arctic Ocean (Aagaard et al., 1987), which then spreads throughout the Arctic Ocean via a cyclonic boundary current and thermohaline intrusion, forming the AWW Layer (Long et al., 2024). In our noMW simulation, even without this advective heat supply, substantial ocean-to-ice heat flux and winter sea ice melt were observed in the Nansen Basin case (such as Figure 8b). This indicates that the winter heat flux from upwelled AWW would be even larger when advective heat input is considered.*

*Moreover, advective transport of sea ice and surface freshwater driven by atmospheric wind patterns can directly influence freshwater sources across different basins of the Arctic Ocean. For example, during the positive Arctic Dipole (AD) phase, freshwater content decreases in the Eurasian Basin and increases in the Amerasian Basin due to the changes of wind-driven Ekman convergence (Wang, 2021). Under the negative Arctic Oscillation (AO) phase, sea ice in the western and central Eurasian Basin exhibits an anticyclonic drift anomaly that accelerates export, while the positive phase reverses this pattern (Wang et al., 2021b). Although our model does not include advective processes, the results effectively reflect their potential impacts. For instance, the noMW run simulates a scenario similar to the net loss of freshwater. In this run, SIMW removal triggers intense vertical mixing in the Nansen Basin (Figure 5r or Figure S11). This implies that in reality, a positive AD phase could produce comparable effects in these regions. Similarly, a positive AO anomaly, which imports sea ice, could enhance regional stratification stability, analogous to the conditions in our noIA run (with sea ice held constant during summer).*

**Q31,** **L419-420 - How does this relate to the observations of Lind et al. (2018) and Skagseth et al. (2020)? Are these examples of conditions that could prevail in the Eurasian basin in the future? How are they captured in the model experiments, please elaborate.**

**Answer:** Thank you very much for this suggestion, which better suits our research within the context of the real Arctic Ocean. In the discussion, we added a paragraph to describe the phenomenon of Atlantic water upward mixing observed by Lind et al.

(2018) and Skagseth et al. (2020) in the Barents Sea, and linked their mechanistic explanations to our experimental results in the Nansen Basin.

**Revision:** Discussion, lines 587-605: *The mechanism by which regional advection-driven sea ice changes affect stratification has been documented in the Barents Sea. Since the mid-2000s, the intense upward mixing of AWW in the Barents Sea has been primarily driven by reduced sea ice inflow (Lind et al., 2018; Skagseth et al., 2020). Sea ice serves as a key freshwater source in this region, maintaining the surface freshwater layer and stable stratification. A reduction in sea ice inflow directly weakens stratification and enhances vertical mixing (Skagseth et al., 2020). This mechanism is also analogous to the findings from our noMW runs in the Nansen Basin, which indicates that, even under current Eurasian Basin conditions, sea ice cover and SIMW release are critical for maintaining stratification and suppressing the upward heat flux from AWW. Many studies have demonstrated that Atlantification is intensifying in the Eurasian Basin (Barton et al., 2018; Muilwijk et al., 2023; Polyakov et al., 2017; Tesi et al., 2021), leading to water properties that increasingly resemble those of the Barents Sea, characterized by warmer temperatures, higher salinity, and weaker stratification. Consequently, as Atlantification advances and Arctic sea ice declines rapidly in the future, the role of SIMW in maintaining stratification will become increasingly important. If sea ice retreats beyond a critical threshold where SIMW production becomes insufficient to sustain stratification and the surface ice cover is too thin to buffer atmosphere-ocean interactions, the basin may experience frequent, intense mixing events. This would eventually lead to significant warming of the entire upper layer, mirroring conditions already observed in the Barents Sea. Therefore, future research should further examine the impact of this atmospheric-ice-ocean coupling on the pace of Arctic Atlantification.*

**Q32, L443 - "well-validated" is subjective, and should be deleted here IMHO.**

**Answer:** Thank you. We deleted this word here and rewrote this part, and also deleted the same word in the Abstract.

**Revision:** Conclusions, lines 686-694: *This study employs a one-dimensional coupled sea ice–ocean model to quantify two key feedbacks under a climatological mean state. Through a series of decoupling experiments, we assessed the independent roles of SIMW and ice-albedo feedbacks in summer melting and their subsequent effects on winter processes. Although our simplified one-dimensional framework does not resolve advective processes or sub-grid scale features, it successfully reproduces the observed evolution of key physical quantities in the central Arctic on seasonal timescales. To the best of our knowledge, this is the first study to quantify the above feedback effects, both individually and in combination, in the Arctic ice-ocean system. Therefore, this work provides a critical conceptual foundation and a quantitative benchmark for understanding these coupled feedbacks. The main conclusions are as follows:*

**Conclusions in general - As noted in the generic comments the results need to be better in context of possible shortcomings I noted in the generic comments, e.g. in relation to the fact you only represent the whole ice cover as level ice? relative to how albedo is treated in the model (and relates to observed albedo), uniform atmospheric forcing vs possibly very regional conditions (esp. in storm tracks in the NS region), and sea-ice meltwater balance in the model vs. observations etc. And what are your recommendations for improving this in future work?**

**Answer:** Thank you again for your comments and many constructive suggestions. Based on your suggestions, we thoroughly addressed in the revised manuscript the limitations of our model setup—such as considering only level ice, neglecting advection, and using climatological atmospheric forcing—and highlighted the differences between these simplifications and real-world conditions. We also emphasized that the objective of this study is to isolate and quantify the strength of the meltwater feedback and the ice-albedo feedback and their interactions under mean-state conditions, so that readers can clearly understand that our conclusions are derived from this mean state analysis.

Regarding the comparison of simulated meltwater volume with observations: in our control experiment, the summer ice melt is approximately 1.1 m (as shown in Fig. R1a), which is equivalent to 1 m of freshwater released to the ocean. This is close to the value of about 1.2 m sea ice meltwater reported by Haine et al. (2015). While our estimate is slightly lower, it is reasonable considering that melt rates in the coastal marginal ice zone are generally higher than those in the central deep basin. As for albedo values, recent studies based on MOSAiC data indicate that the observed albedo ranges from approximately 0.55 to 0.64 across thin ice (less than 0.5 m) to thick ice (greater than 1 m), with relatively stable values for ice thicker than 1 m (Light et al., 2022). In our simulations, summer sea ice thins from 2.0 m to 0.9 m, accompanied by a decrease in albedo from 0.63 to 0.58. These results suggest that our simulated albedo values are in the range of observations. We included the comparisons of both meltwater volume and albedo with observational data in the "Model Validation" section of the Supplementary Information and mentioned in the main text.

Furthermore, in the main text, and in light of the discussion on model limitations, we also added a paragraph in the Discussion about our recommendations for improving this work in the future.

**Revision:** Section 2.4, lines 282-286: *We evaluated this 1D model against observations in several aspects, including the seasonal variation of vertical temperature-salinity structure, the volume of meltwater release, ice-ocean heat flux, and ice-albedo values. The results demonstrate that this simplified model can replicate the observed seasonal cycles of these key physical variables in the ice-ocean system well (see Section 2.4 in the SI).*

**Revision:** Discussion, lines 675-684: *The primary contribution of this study lies in clearly demonstrating the independent and interactive effects of SIMW and ice-albedo feedbacks on the ice-ocean system under mean-state conditions, thereby providing a useful conceptual framework for future quantification of other coupled feedback mechanisms. We suggest that future research should build upon this foundation by integrating three-dimensional models to incorporate advective processes; employing*

*high-resolution forcing to resolve storm-induced variability and transient events; refining the representation of sub-ice topography and melt pond effects to improve feedback estimates. Additionally, extending investigations to inter-annual timescales and incorporating winter-specific feedbacks (e.g., ice production-entrainment and growth-thickness mechanisms) will be crucial for projecting the changes of Arctic Atlantification and sea ice loss under evolving climate forcings.*

**Q33,** **Other relevant literature that might be useful to include** 'Van Straaten, C., Lique, C., & Kolodziejcyk, N. (2025). The Life Cycle of the Low Salinity Lenses at the Surface of the Arctic Ocean. Journal of Geophysical Research: Oceans, 130(4), e2024JC021699. https://doi.org/10.1029/2024JC021699'

**Answer:** In addition to the other references you provided, we found this work regarding the impact of the low-salinity lenses in shelf regions is interesting. Therefore, we cited it in the discussion with the comparison with our results.

**Revision:** Discussion, lines 654-662: *A recent study revealed the prevalence of low-salinity lenses in the marginal ice zones and shallow shelves of the Arctic Ocean, formed by localized intense sea-ice melt and river runoff. These lenses enhance summer sea-ice melting by trapping and concentrating solar radiation near the sea surface (Van Straaten et al., 2025). This effect contrasts sharply with the melt-inhibiting role of the SIMW in the deep basin identified in our study. The divergence underscores that the influence of surface freshwater layers on sea-ice cover differs fundamentally between shallow shelves and the deep basin. As sea ice retreat accelerates and the marginal ice zone shifts further into the central Arctic Ocean, the melt-enhancing effect of such lenses is expected to grow in importance. Therefore, the pan-Arctic integrated effect and regional variability of meltwater, or surface freshwater layers more broadly, warrant further investigation*.

**Response to Reviewer#2**

**General Comments:**

The overall objective of the study is to *quantify the independent effects of meltwater and ice-albedo feedbacks on the Arctic ice-ocean system*. Specifically, the work proposes to answer three questions:

- *To what extent do the meltwater and ice-albedo feedbacks influence ice melting during summer?*
- *What is the contribution of the meltwater (ice-albedo) feedback if the ice-albedo (meltwater) feedback is not involved?*
- *What are the regional differences in the impact of meltwater on the ice-ocean system?*

These research objectives are within the scope of The Cryosphere and, to the best of my knowledge, are novel. The difficulty of the research arises from the strong coupling between the feedbacks of meltwater and ice albedo, which prevents a clear differentiation of the contribution of each. To solve this problem, the authors propose a methodology based on a series of simulations where feedbacks are selectively activated. Independent contributions are then encoded in feedback factors. I consider the methodology appropriate, implemented in a rigorous way and well documented. The supplementary material reinforces its validity with an extensive comparison between the model and observations. However, I believe that its description would benefit from the inclusion of the 1D dynamic equations either in the main text or in the supplementary material.

The presentation and discussion of the results raised my main concerns with the manuscript. Model results are quantitatively presented but, in main opinion, poorly interpreted from the physical point of view. In some sections (see below), the explanation of some model results would require a qualitative interpretation or a physical hypothesis. The quantities are only indicated and at most justified with references or general assessments. Brief physical explanations would be very informative.

I consider that a revised version of the manuscript could be suitable for publication in The Cryosphere.

**Answer:** Thank you very much for your detailed review and constructive suggestions, which significantly improved the quality and readability of our manuscript. Based on your comments and suggestions, we made the following key revisions: Provided specific details about the equations about the ocean model, sea ice model, and albedo parameterization in the Supplementary Information; Optimized some pictures both in the main text and supplementary information; Further refined the Abstract and Conclusion sections; Restructured the section of Model Results. Originally, the section of 'sea ice response' were described before 'ocean response', this order has now been reversed to more logically describe how oceanic processes drive the observed sea ice variability. We now answer each of your comments point by point below. For clarity, **your comments are in bold black**, our responses are in blue, and *the revised text in*

*the manuscript is presented in italic*. All authors approved these changes. In this response, figures are labeled as Fig. R1, Fig. R2, etc.

**Specific Comments:**

**Q1, L55:** *In winter, brine rejection caused by ice formation leads to the upward entrainment of heat from the NSTM and AWW and impedes winter ice formation consequently (Smith et al., 2018; Steele et al., 2011; Timmermans et al., 2017), which is negative feedback known as the ice production-entrainment feedback (Goosse et al., 2018).* **My understanding is that the ice production-entrainment feedback inhibits vertical heat transfer, promoting more ice production. could you confirm this point?**

**Answer:** Thank you for your comment. Indeed, the "production-entrainment feedback" refers to the process whereby sea ice formation releases brine, increasing surface salinity. This process can enhance vertical convection and mixed layer deepening. As a result, heat from deeper ocean layers (such as the Near-Surface Temperature Maximum and Atlantic Warm Water) is entrained upward, thereby delaying further ice formation. We acknowledge the lack of a clear explanation of this mechanism in the original manuscript. We revised the relevant text in the Introduction to clarify the production-entrainment feedback.

**Revision:** Introduction, lines 65-74: *In winter, the upward mixing of heat from subsurface and deep layers to mixed layer (e.g., the NSTM and AWW) can significantly impede sea ice growth (Smith et al., 2018; Steele et al., 2011; Timmermans et al., 2017). In particular, brine rejection during sea ice formation is an important driver of this upward heat entrainment: the release of dense, saline water caused by sea water freezing enhances vertical convection, deepens the mixed layer, and draws oceanic heat into the surface layer (Polyakov et al., 2020; Zhong et al., 2024). This heat flux usually does not stop ice formation, otherwise there would be no convective mixing and ongoing heat release to the mixed layer; instead, this heat flux represents a negative feedback mechanism that reduces the rate of winter ice growth (Polyakov et al., 2013), known as the ice production-entrainment feedback (Goosse et al., 2018).*

**Q2, L 72** *Zhang et al., (2023) demonstrated that the removal of meltwater increases ice melt by 17%* **Could you add briefly the physical mechanism proposed by Zhang et al. to justify this effect?**

**Answer:** Thank you for your comment. We added a sentence here to justify this effect. **Note:** Based on another reviewer's suggestion, we clarified the term "meltwater" as "sea-ice meltwater (SIMW)" throughout the text, as it appears frequently, hence this explanation.

**Revision:** Introduction, lines 97-100: *Zhang et al. (2023) demonstrated that the removal of SIMW in a coupled sea ice-ocean model increases ice melt by 17%. As meltwater removal weakens ocean stratification, allowing more heat from NSTM to reach the ice base, thereby accelerating melting.*

**Q3, L 75 Indicate the differentiated aspects considered in the work between meltwater and ice -albedo feedbacks. There is a strong relationship between both and it is not clear their intrinsic and differentiated components.**

**Answer:** Thank you for this constructive suggestion, which really helps enhance the significance of our study.

**Revision:** Introduction, lines 88-94: *However, the SIMW and ice-albedo feedbacks, while strongly coupled, exhibit distinct physical mechanisms and impacts: the SIMW feedback primarily arises from the freshwater input during sea ice melting, which enhances ocean stratification and decreases the ocean-to-ice heat flux, and can suppress further ice melting (a negative feedback). This process also influences winter ice formation by modulating heat entrainment (Zhang et al., 2023). In contrast, the ice-albedo feedback stems from the reduction in sea ice cover, which lowers surface albedo, increases solar radiation absorption, and amplifies ice melting (a positive feedback).*

**Q4, L 102 I would suggest to include the mathematical expression of the 1D model.**

**Answer:** Thank you for this constructive suggestion. We agree that including the 1D dynamic equations will improve the clarity and reproducibility of the model description. Accordingly, we added a new section in the Supplementary Information, which presents the key 1D equations for the ocean and sea ice components, as well as the Albedo parameterization.

**Q5, L 116: A boundary condition is always required to get a particular solution to the differential equation. Please, clarify the sentence *No boundary condition is applied at the bottom of the model*.**

**Answer:** Thank you for this insightful comment. Sorry for the miswording in our original phrasing. The sentence in the first manuscript "No boundary condition is applied at the bottom of the model" was intended to mean that we applied a closed (no-flux) boundary condition (i.e., a closed boundary with zero normal velocity and no material flux). We revised the relevant sentence in the Methods section to explicitly state.

**Revision:** Section 2.1, lines 151-152: *At the bottom, a closed boundary condition is applied, meaning that there is no exchange of mass, momentum, or other properties across this boundary.*

**Q6, L 175 *Based* ON *the feedback factor (γ) framework proposed by Goosse…***

**Answer:** Thank you for your careful review. We revised this sentence.

**Q7, L 177 *Total response* by Total Response**

**Answer:** Thank you for your careful review. We revised this sentence.

**Q8,** L 179 (Reference Response, RR)

**Answer:** Thank you for your careful review. We revised this sentence.

**Q9,** Figure 3. Please define the acronyms in the figure caption. The figure is cited before they are defined in the text.

**Answer:** Thank you for your careful review. We revised this figure caption.

**Revision:** Section 2.4, lines 295-303: *Figure 3. Schematic representation of the four experiments in this study. (a): CTRL (Control experiment), ice-ocean coupled model running normally; (b): noMW (no meltwater feedback experiment), after sea ice melting, the sea ice meltwater flux entering the ocean is set to 0; (c): noIA (no ice-albedo feedback experiment), after sea ice melting, sea ice meltwater enters the ocean normally, but before the model starts the next time step, sea ice is restored to its first melt-day state; (d): noMWIA (no ice-albedo feedback and meltwater feedback experiment), after sea ice melts, the sea ice meltwater flux entering the ocean is set to 0, and simultaneously, before the model starts the next time step, the sea ice is restored to its first melt-day state. The red arrows represent the shortwave radiation fluxes: SWice (Shortwave radiation reaching the ice surface), SWreflect (shortwave radiation reflected from the ice surface), SWtransmitted (shortwave radiation transmitted through sea ice to the ocean), and SWopen-ocean (shortwave radiation entering the ocean through open ocean).*

**Q10,** L 264-265 *the brine rejection process during winter creates higher ocean-ice heat flux, which in turn inhibits ice formation.* **Could you briefly clarify the mechanism of this? As I understand it, brine rejection contributes to the formation of the cold halocline that isolates the upper Arctic layers from the warm waters of the Atlantic. It would favor ice formation in this case. Could you clarify your assement?**

**Answer:** Thank you for your comment. The process of brine rejection is indeed one of the key processes contributing to the formation of the Arctic halocline. However, this process primarily occurs through ventilation from shallow shelf regions to the deep basin—that is, the dense, saline water formed during sea ice formation in shallow areas flows along isopycnal surfaces into the halocline layer of the deep basin, helping to sustain it (as illustrated in Figure R1 below, adapted from Metzner et al., 2020). Locally, brine rejection tends to enhance vertical convection in the upper ocean. We acknowledge that this point was not clearly explained in our original manuscript. In the revised version, we added several sentences to better clarify this mechanism.

[Figure]

**Figure. R1.** Schematic of the formation of the cold halocline and vertical stratification of the Arctic Ocean (cited from Metzner et al., 2020).

**Revision:** Section 3.2.2, lines 442-459: *During the freezing season, sea ice evolution can be influenced by the stratification of the initial ocean profile (Figure 7b). In the CTRL run, stations NS2-NS5 with weak stratification exhibit less ice formation in winter compared to strongly stratified regions (Figure 7b, CTRL). This difference is governed by how ocean stratification modulates vertical heat transport. In strongly stratified regions, a pronounced halocline acts as a stable barrier. This halocline is primarily formed and sustained by a ventilation process: intense freezing in seasonal ice zones over the shelf seas generates cold, dense, and saline waters, which flow into the deep basins along isopycnal surfaces and feed the halocline (Itkin et al., 2015; Metzner et al., 2020). Because of this pre-existing, stable layer with a strong salinity gradient (e.g., Figure 2a), local wintertime brine rejection is insufficient to erode it, making it difficult to bring deep heat to the surface layer, thereby promoting ice formation. Conversely, at stations NS2-NS5 where stratification is weak, the halocline is shallow and unstable, and the warm AW is closer to the surface (e.g., Figure 2j and k). Here, vertical convection induced by local brine rejection readily entrains heat from the underlying AW. The resultant increase in the ocean-to-ice heat flux thereby suppresses sea ice formation. This mechanism is consistent with Linders and Björk, (2013), who demonstrated that weak salinity stratification allows winter ocean heat fluxes of up to 8 W m-2 to reach the ice, whereas a well-developed halocline reduces fluxes to ~0.7 W m-2. Similarly, Polyakov et al., (2013) observed that winter convection in the eastern Arctic Ocean transports heat from AWW to the surface layer, reducing ice formation in weakly stratified areas.*

**Q11,** **L 270** *the removal of meltwater results in a 0.19 m increase in ice formation compared to the CTRL run.* **Why? How do you physically interpret this result?**

**Answer:** Thank you for your comment. We fully agree that a clear physical explanation is necessary here. In the experiment with meltwater removal, the increase in winter ice formation is primarily due to enhanced sea ice melting during summer, which results in thinner sea ice and a larger open water area by the time winter arrives. This leads to two key effects: (1) Thinner ice exhibits higher thermal conductivity, causing faster heat loss at

the ice base (Figure R2c and d below); and (2) The expanded open water area facilitates more rapid ocean heat loss (Figure R2e and f below). Consequently, ice formation in early winter is increased in the noMW run. We added the physical explanation in the revised manuscript and included a time series plot of heat flux caused by thermal conduction from the ice surface to base ($F_{cb}$, its equation is also provided in the Supplementary Information) in Figure 8 (as shown in the Figure R2 below) to help illustrate this physical interpretation.

[Figure]

**Figure R2.** Heat flux time series at station BS5 (left column) and NS2 (right column). (a)-(b): $F_{oi}$ (ocean-to-ice heat flux); (c)-(d): $F_{cb}$ (heat flux caused by thermal conduction from the ice surface to base); (e)-(f): $F_{oa}$ (ocean-atmosphere heat flux over the open ocean) for the experiments with initial SIT of 2 m. In the (a)-(d), positive (negative) values denote heat gain (loss) at the ice base. In the (e)-(f), positive (negative) values denote upward (downward) heat flux, corresponding to oceanic heat loss (gain).

**Revision:** Abstract, lines 19-21: *……intensified summer ice melt enhances early winter ice formation in strongly stratified regions, as reduced ice thickness and expanded open water areas in early winter accelerate oceanic heat loss, thereby promoting rapid freezing.*

**Revision:** Section 3.2.2, lines 464-473: *This is because, enhanced summer ice melt in the noMW run results in thinner sea ice and more extensive open water by the start of winter (Figure 4a and c). These conditions favor rapid ice formation in early winter through two key mechanisms. First, the thinner ice allows more efficient thermal conduction, speeding up ice basal cooling during early winter (Figure 8c). Second, the larger areas of open-ocean promote stronger heat loss from the ocean to the atmosphere, as seen in the elevated ocean-to-atmosphere heat fluxes in the noMW run (Figure 8e). The synergy of these processes drives faster ice growth, demonstrating that summer ice retreat can paradoxically enhance early winter ice formation by improving both conductive and surface heat loss. This phenomenon is consistent with findings from previous studies. Analyses of CMIP6 model data and ice mass balance buoys indicate that basal growth increases in winter due to thinner sea ice during the 21st century (Keen et al., 2021; Lin et al., 2022).*

**Q12,** L 273-275 *The noMWIA run also shows less ice formation compared to the noIA run (Figure 5b) in station NS2, NS4 and NS5, but no sea ice melting is observed during winter, further demonstrating the importance of meltwater in weakly stratified regions.* **What is the physical explanation for this result?**

**Answer:** Thank you for your comment. The physical mechanism is similar to what we described in response to Q11. In the noMWIA experiment, although meltwater was also removed, we simultaneously maintained the summer sea ice concentration and thickness unchanged. This leads to the thick sea ice and very small open water area by the time winter arrives. As a result, heat loss at the ice base and from the ocean are slow (as shown in Figures R2c and e above), resulting in slower ice formation in the noMWIA run. Consequently, the convection driven by brine rejection is weaker and insufficient to disrupt the pre-existing stratification. Thus, unlike in the noMW run (where only meltwater was removed), no intense vertical mixing occurs in the noMWIA run. We added an explanation of the physical mechanism in the revised manuscript.

**Revision:** Section 3.2.2, lines 489-500: *The influence of SIMW is further demonstrated by comparing the noMWIA and noIA experiments. As shown in Figure 5b, the noMWIA run exhibits reduced sea ice formation in the NS2, NS4, and NS5 regions relative to the noIA run. In the absence of SIMW, weakened stratification intensifies winter convection and upward heat transport, increasing the ocean-to-ice heat flux (Figure 8b) and thereby slowing ice growth in these regions (Figure 4b). However, unlike the noMW run where strong winter vertical mixing and sea ice melt are observed (Figure 4b and Figure 5r), these phenomena do not occur in the noMWIA run. This discrepancy arises because the noMWIA experiment maintains summer sea ice cover unchanged, leading to thicker sea ice and a much smaller open ocean by the onset of winter. Consequently, both ice basal and oceanic heat loss proceed slowly (Figure 8c–f), reducing the winter ice formation. The associated weakening of brine rejection further weakens the ocean vertical convection, which becomes insufficient to disrupt the pre-existing stratification. As a result, AWW cannot reach the surface (Figure 5t), and winter ice melt does not occur (Figure 4b) in the noMWIA run.*

**Q13,** L 277 *This suggests that sea ice retreat during summer can promote ice formation in winter.* **Can you give a physical reason based on your model? Is it just a geometric effect of having more ocean surface available to freeze.**

**Answer:** Thank you for your comment. As we responded in the **Q11** comment, this is not only due to the geometric effect of having more ocean surface available to freeze (as illustrated in Figures R2e or f above, comparing the red and blue lines), but also because thinner sea ice exhibits higher thermal conductivity. We added some summary statements to this section to provide further clarification. Please refer to our Answer and Revision in **Q11**.

**Q14,** L 291 the *NSTM, because the heat absorbed by the ocean can mix sufficiently within the upper 30 m*. **This explanation would disagree with observations. The NSTM layer can be observed at this depth in the Canada Basin (Jackson 2010). Provide an explanation to the discrepancy between the model and observations.**

**Answer:** Thank you for your comment. We conducted a more detailed investigation into the literature regarding the depth of the NSTM based on your comment. The study by Jackson et al. (2010) indeed indicates that the NSTM can occur at depths greater than 30 meters, but this phenomenon was mainly observed in the 1990s. After 2000, the depth of the NSTM has generally been around 20 meters (e.g., Table 3 below, cited from Jackson et al., 2010). Gallaher et al. (2017), based on observational results, also show that the depth of the NSTM is around 10–25 meters. Their findings further indicate a significant shoaling trend in the NSTM from the late 20th century to the early 21st century. Our control experiment simulates the NSTM at depths of approximately 10–25 meters, which aligns well with the observations. Additionally, we acknowledge that our explanation here was not sufficiently clear. For a clearer presentation of the results, we revised the relevant sections in the manuscript. In the Introduction, we added a sentence to introduce the depth of the NSTM. In Section 3, we rewritten the sentence about describing the results and comparisons between the Control run and noMW run.

**Table 3.** Comparison of the Average Depth, With Standard Error, of the NSTM and the Average Depth of the Maximum Brunt-Väisälä Frequency in the Northern Canada Basin During the Summers of 1993, 1997, and 2002–2007[a]

| Year | Depth of NSTM (m) | Depth of Maximum Brunt-Väisälä Frequency (m) |
|---|---|---|
| 1993 | 26.8 ± 6 | 27.6 ± 4 |
| 1997 | 35.0 ± 1 | 33.0 ± 1 |
| 2002 | 1 | 28.0 ± 4 |
| 2003 | 16.6 ± 3 | 33.5 ± 4 |
| 2004 | 21.3 ± 1 | 18.8 ± 8 |
| 2005 | 18.2 ± 3 | 17.4 ± 3 |
| 2006 | 20.9 ± 1 | 18.6 ± 1 |
| 2007 | 13.6 ± 2 | 16.3 ± 4 |

[a]Here, only stations that were north of 75°N and had a bottom depth greater than 3500 m were used to calculate the average. For standard error, the value of n was 3 in 1993, 2 in 1997, 3 in 2002, 12 in 2003, 13 in 2004, 11 in 2005, 13 in 2006, and 17 in 2007. In 2002, only one station had an NSTM that was 1 m deep.

**Revision:** Introduction, lines 62-65: *Observations in the Canada Basin have revealed a significant shoaling trend of the NSTM, with its mean depth shoaling from more than 30 m in the late 1990s to approximately 20 m in the 21st century, largely driven by enhanced surface stratification resulting from accelerated sea ice melt (Gallaher et al., 2017; Jackson et al., 2010; Steele et al., 2011).*

**Revision:** Section 3.1.1, lines 330-335: *The NSTM forms because the low-salinity SIMW layer strengthens the vertical density gradient, limiting convective heat loss and allowing it to persist as a residual warm layer at the base of the mixed layer (Alvarez, 2023; Jackson et al., 2010; Steele et al., 2011). In our simulation, the NSTM layer is typically observed at depths of approximately 10–25 m (Figure 5a), consistent with observed NSTM depths (Gallaher et al., 2017; Jackson et al., 2010). Additional information on model validation can be found in Section 2.4 and Figure S7 in the SI.*

**Revision:** Section 3.1.1, lines 339-346: *A prominent result is the disappearance of the NSTM in the noMW and noMWIA runs (Figure 5b and d). For example, at station BS5 (strongly stratified), SIMW removal increases the summer MLD to approximately 30 m (Figure 5b), which prevents NSTM formation because, in the absence of SIMW-induced summer shallow mixed layer, the heat absorbed by the ocean mixes thoroughly within the upper 30 m, resulting in a uniform temperature profile (Figure 5b), rather than a warm layer isolated at the mixed layer base (Figure 5a). This results in the heat that could originally be stored in the NSTM at the bottom of the summer shallow mixed layer directly contacting the ice bottom, causing more sea ice melting......*

**Q15,** L 297 ...agree well with the observed values...

**Answer:** Thank you for your careful review. We revised this sentence.

**Q16,** Figure 6. Add month labels to the figures corresponding to station BS5, similarly to those of NS2.

**Answer:** Thank you for your careful review. We revised this figure, and we also incorporated the results from the station AM2 station as suggested by another reviewer (as shown in Figure R3 below).

[Figure]

**Figure R3.** Simulated vertical profiles of ocean temperature and salinity over time at stations (a)-(h) BS5, (i)-(p) AM2, and (q)-(x) NS2. For each station, the top row displays temperature, and the bottom row displays salinity. From left to right, the columns correspond to the CTRL run, noMW run, noIA run, and noIAMW run. Cyan dots in each panel represent the mixed layer depth. Note that the vertical depth scales are not consistent across stations.

**Q17,** Figures 6 a,e,I,m,c,g,k and o. The discontinuity in the MLD seems an artifact of the surface tracking algorith due to the appearance of a new water mass, the meltwater. A gradient tracking algorithm could identify this as the bottom of the mixed layer. There is no discontinuity in the rest of simulations because the meltwater input is cancelled. Could you clarify this point?

**Answer:** Thank you for your comment. We fully agree that the discontinuity of the MLD is caused by the sea-ice meltwater. In fact, both the density threshold method and the density gradient tracking method can produce this discontinuity (see Figure R4 below). The main reason is that when sea-ice meltwater input begins, a fresh surface layer forms. This causes a sharp change in surface reference values like density or salinity. As a result, the MLD becomes much shallower. As you noted, this jump only occurs in simulations with meltwater, CTRL and noIA run. In experiments without meltwater (noMW and noMWIA), the MLD changes more smoothly. In addition, our model uses a 600-second time step and saves data daily. Daily sampling captures the state after meltwater stratification occurs. This makes MLD jumps appear as discrete daily changes in the plots, which may amplify the MLD discontinuity. We added some sentences in the revised manuscript to clarify the simulated MLD discontinuity caused by the sea-ice meltwater input.

[Figure]

**Figure R4.** Mixed layer depth at the BS station, calculated using the density threshold method and the density gradient tracking method. Background color indicates the vertical density gradient. (a) CTRLrun; (b) noIA run.

**Revision:** Section 3.1.1, lines 352-357: *It should be noted that in the CTRL and noIA runs, the MLD shows a discontinuity during the early melting season. This occurs due to the input of SIMW into the ocean. Specifically, when sea ice begins to melt, the buoyancy flux term changes sign. This change leads to the formation of a new, shallower, and fresher surface layer. As a result, the MLD shoals rapidly. In contrast, the noMW and noMWIA runs do not exhibit this discontinuity. This is because SIMW is removed in these two experiments.*

**Q18,** L 305 How do you explain this big impact of meltwater in the mixed layer depth during the freezing season? Is there any observational or 3D model result that corroborates it?

**Answer:** Thank you for your comment. This big impact can be attributed to the combined effect of the initially weak stratification in these regions and the removal of meltwater. When meltwater is removed, the surface loses a portion of its freshwater source. As winter sets in, convection driven by brine rejection during ice formation more readily disrupts this already weak oceanic stratification. Although we not found direct observational evidence confirming such an effect of meltwater in this specific area, observational studies have documented upward mixing of Atlantic Water caused by brine rejection convection (Polyakov et al., 2017, 2020). Additionally, research has shown that in the Barents Sea, reduced sea ice inflow has decreased surface meltwater sources, leading to pronounced upward mixing of Atlantic Water and warm in the upper ocean (Lind et al., 2018; Skagseth et al., 2020).

   Our study actually highlights that meltwater already plays a significant role in the Nansen Basin. In the future, with increasing Atlantification in the Eurasian Basin, the sea waters are expected to become warmer and saltier, approaching a state similar to that of the Barents Sea. Consequently, the upper-ocean warming mechanism observed in the Barents Sea may likely recur in the Nansen Basin and even across the Eurasian Basin. We revised the relevant sections in the manuscript to include a comparative discussion between our findings and observational studies. We greatly appreciate this constructive comment.

**Revision:** Section 3.1.2, lines 366-377: *The pronounced impact of SIMW on MLD during the freezing season at stations NS2–NS5 can be attributed to the interplay between pre-existing weak stratification and the absence of SIMW-induced freshwater input. These stations inherently exhibit weak stratification (Figure 2l). In the noMW run, the removal of summer SIMW results in higher surface salinity and further weakens stratification compared to the CTRL run (Figure 5u and v). As winter begins, sea ice formation drives brine rejection and associated convection. In the noMW run, the lack of residual SIMW leads to stronger brine rejection convection, which eventually overwhelms the already weak stratification, leading to intense vertical mixing and a rapid deepening of the MLD. In contrast, in the CTRL run, the presence of surface SIMW maintains sufficient stratification to resist the convection induced by freezing, thereby limiting mixed layer deepening. Observations have shown that in the Nansen basin with weak stratification, winter brine rejection can drive intense vertical mixing and upward heat flux from the AWW layer, significantly impacting sea-ice growth (Polyakov et al., 2017, 2020).*

**Revision:** Discussion, lines 587-605: *The mechanism by which regional advection-driven sea ice changes affect stratification has been documented in the Barents Sea. Since the mid-2000s, the intense upward mixing of AWW in the Barents Sea has been primarily driven by reduced sea ice inflow (Lind et al., 2018; Skagseth et al., 2020). Sea ice serves as a key freshwater source in this region, maintaining the surface*

*freshwater layer and stable stratification. A reduction in sea ice inflow directly weakens stratification and enhances vertical mixing (Skagseth et al., 2020). This mechanism is also analogous to the findings from our noMW runs in the Nansen Basin, which indicates that, even under current Eurasian Basin conditions, sea ice cover and SIMW release are critical for maintaining stratification and suppressing the upward heat flux from AWW. Many studies have demonstrated that Atlantification is intensifying in the Eurasian Basin (Barton et al., 2018; Muilwijk et al., 2023; Polyakov et al., 2017; Tesi et al., 2021), leading to water properties that increasingly resemble those of the Barents Sea, characterized by warmer temperatures, higher salinity, and weaker stratification. Consequently, as Atlantification advances and Arctic sea ice declines rapidly in the future, the role of SIMW in maintaining stratification will become increasingly important. If sea ice retreats beyond a critical threshold where SIMW production becomes insufficient to sustain stratification and the surface ice cover is too thin to buffer atmosphere-ocean interactions, the basin may experience frequent, intense mixing events. This would eventually lead to significant warming of the entire upper layer, mirroring conditions already observed in the Barents Sea. Therefore, future research should further examine the impact of this atmospheric-ice-ocean coupling on the pace of Arctic Atlantification.*

**Q19,** L 390 *This is a process similar to the noMW run, where less meltwater entering the ocean accelerates sea ice melting.* **Meltwater from ponds, sooner or later, flows into the Ocean. This might slow or delay the formation of a meltwater layer, but probably not inhibit it like in the study.**

**Answer:** Thank you for your comment. We agree that the melt ponds only retain a portion of meltwater on the sea ice surface. Accordingly, we revised the relevant sentence to clarify that the primary impact of melt ponds is through altering surface albedo, rather than reducing meltwater input.

**Revision:** Discussion, lines 640-653: *In the Arctic Ocean, melt pond formation is a process associated with SIMW. The melt ponds develop from the accumulation of snowmelt and surface sea ice melt, with their spatiotemporal morphology controlled by surface topography (Petrich et al., 2012; Polashenski et al., 2012; Webster et al., 2015). Melt ponds can temporarily retain SIMW, delaying its drainage into the ocean, a behavior somewhat analogous to the idealized "noMW" experimental scenario, which entirely prevents SIMW drainage to quantify its feedback but represents an extreme assumption. Observational evidence indicates that only about 10–15% of SIMW is retained on the ice surface in reality (Perovich et al., 2021; Smith et al., 2025). Zhang et al. (2023) demonstrated that removing 20% of SIMW input results in only a 1% increase in summer sea ice melt, implying that the direct influence of SIMW retention caused by the ponds on ice cover is limited. In contrast, the albedo effect associated with melt ponds likely plays a more substantial role in sea ice melt. Melt ponds act as "windows" for solar radiation, significantly reducing surface albedo and enhancing radiation transmission into the upper ocean (Nicolaus et al., 2012), thereby accelerating ice melt. Consequently, future research can investigate more the interplay between SIMW retention and albedo effects to fully understand their combined impact on Arctic sea ice loss.*

**Q20,** L 455 *it delays winter ice melting by slowing surface cooling.* **Should be warming instead cooling?**

**Answer:** Thank you for your careful review. We rewrote this sentence to correct the wording error and improve clarity.

**Revision:** Conclusions, lines 703-707: *There is a seasonal compensation mechanism for sea ice: enhanced summer ice melt can, paradoxically, lead to increased ice formation during the subsequent winter in strongly stratified regions. This occurs because greater summer melt results in a thinner ice cover and expanded open water areas by the start of winter, which collectively enhance the ocean heat loss to the atmosphere and accelerate the freeze-up in the early winter.*